# *Drosophila* SUMM4 complex couples insulator function and DNA replication control

**Evgeniya N Andreyeva[1†], Alexander V Emelyanov[1†], Markus Nevil[2], Lu Sun[3], Elena Vershilova[1], Christina A Hill[4], Michael-C Keogh[3], Robert J Duronio[4,5,6,7], Arthur I Skoultchi[1], Dmitry V Fyodorov[1]\***

[1]Department of Cell Biology, Albert Einstein College of Medicine, Bronx, United States; [2]UNC-SPIRE, University of North Carolina, Chapel Hill, United States; [3]EpiCypher, Durham, United States; [4]Integrative Program for Biological and Genome Sciences, University of North Carolina at Chapel Hill, Chapel Hill, United States; [5]Lineberger Comprehensive Cancer Center, University of North Carolina, Chapel Hill, United States; [6]Department of Biology, University of North Carolina, Chapel Hill, United States; [7]Department of Genetics, University of North Carolina, Chapel Hill, United States

**\*For correspondence:**
dmitry.fyodorov@einsteinmed.org

[†]These authors contributed equally to this work

**Abstract** Asynchronous replication of chromosome domains during S phase is essential for eukaryotic genome function, but the mechanisms establishing which domains replicate early versus late in different cell types remain incompletely understood. Intercalary heterochromatin domains replicate very late in both diploid chromosomes of dividing cells and in endoreplicating polytene chromosomes where they are also underreplicated. *Drosophila* SNF2-related factor SUUR imparts locus-specific underreplication of polytene chromosomes. SUUR negatively regulates DNA replication fork progression; however, its mechanism of action remains obscure. Here, we developed a novel method termed MS-Enabled Rapid protein Complex Identification (MERCI) to isolate a stable stoichiometric native complex SUMM4 that comprises SUUR and a chromatin boundary protein Mod(Mdg4)-67.2. Mod(Mdg4) stimulates SUUR ATPase activity and is required for a normal spatio-temporal distribution of SUUR in vivo. SUUR and Mod(Mdg4)-67.2 together mediate the activities of *gypsy* insulator that prevent certain enhancer–promoter interactions and establish euchromatin–heterochromatin barriers in the genome. Furthermore, *SuUR* or *mod(mdg4)* mutations reverse underreplication of intercalary heterochromatin. Thus, SUMM4 can impart late replication of intercalary heterochromatin by attenuating the progression of replication forks through euchromatin/heterochromatin boundaries. Our findings implicate a SNF2 family ATP-dependent motor protein SUUR in the insulator function, reveal that DNA replication can be delayed by a chromatin barrier, and uncover a critical role for architectural proteins in replication control. They suggest a mechanism for the establishment of late replication that does not depend on an asynchronous firing of late replication origins.

## Editor's evaluation

This important paper will be of interest to those studying DNA replication in the context of chromatin and development and to those interested in higher-order chromatin organization. It uncovers a new interaction partner for SuUR and reports how this complex (SUMM4; Suppressor of Underreplication – Modifier of Mdg4) functions to control under-replication. The results are convincing and support the conclusions.

**eLife digest** Inside cells, molecules of DNA provide the instructions needed to make proteins. Cells carefully maintain and repair their DNA, and typically make a complete copy of the genome before they divide to ensure that after division, each daughter cell has a full set.

Within human, fly and other eukaryotic nuclei, DNA is packaged into structures known as chromosomes. Cells follow precisely controlled programs to replicate distinct regions of chromosomes at different times. To start copying a particular region, the cell machinery that replicates DNA binds to a sequence known as the origin of replication. It is thought that as-yet unknown cues from the cell may lead the replication machinery to bind to different origins of replication at different times.

In some circumstances, cells make extra copies of their DNA without dividing. For example, many cells in the larvae of fruit flies contain hundreds of extra DNA copies to sustain their increased sizes. However, the entire genome is not copied during this process, so cells end up with more copies of some regions of the genome than others. A protein called SUUR is required for hindering the replication of the 'underrepresented' regions, but it is not clear how it works.

To address this question, Andreyeva, Emelyanov et al. developed a new approach based on liquid chromatography and quantitative proteomics to identify the native form of SUUR in fruit flies. This revealed that SUUR exists as a stable complex with a protein called Mod(Mdg4), which is needed to recruit SUUR to the chromosomes. Further experiments suggested that SUUR and Mod(Mdg4) work together to bind to regions of DNA known as *gypsy* insulator elements, creating a physical barrier that hinders the replication machinery from accessing some parts of the genome.

The findings of Andreyeva, Emelyanov et al. provide an alternative explanation for how individual cells may stagger the process of copying their DNA without relying on the replication machinery binding to various replication origins at different times. Rather, late replication timing may be instructed by an insulator-born delay of the progression of replication over particular genomic regions. This mechanism adds to the list of nuclear processes (chromosome partitioning, transcriptional regulation, etc.) that are known to be directed by insulators and associated architectural proteins.

## Introduction

Replication of metazoan genomes occurs according to a highly coordinated spatiotemporal program, where discrete chromosomal regions replicate at distinct times during S phase (*Rhind and Gilbert, 2013*). The replication program follows the spatial organization of the genome in Megabase-long constant timing regions interspersed by timing transition regions (*Marchal et al., 2019*). The spatio-temporal replication program exhibits correlations with genetic activity, epigenetic marks, and features of 3D genome architecture and subnuclear localization. Yet the reasons for these correlations remain obscure. Interestingly, the timing of firing for any individual origin of replication is established during G1 before pre-replicative complexes (pre-RC) are assembled at origins (*Dimitrova and Gilbert, 1999*), suggesting a mechanism that involves factors other than the core replication machinery.

Most larval tissues of *Drosophila melanogaster* grow via G-S endoreplication cycles that duplicate DNA without cell division, resulting in polyploidy (*Zielke et al., 2013*). Endoreplicated DNA molecules frequently align in register to form giant polytene chromosomes (*Zhimulev et al., 2004*). Importantly, in some cell types, genomic domains corresponding to the latest replicated regions of dividing cells, specifically pericentric (PH) and intercalary (IH) heterochromatin, fail to fully replicate during each endocycle resulting in underreplication (UR). These regions are depleted of sites for binding the Origin of Replication Complex (ORC), and thus, their replication primarily relies on forks progressing from external origins (*Sher et al., 2012*) in both dividing and endoreplicating cells, which suggests that both cell types utilize related mechanisms of regulation of late replication. Although cell cycle programs are dissimilar between endoreplicating and mitotically dividing cells (*Zielke et al., 2013*), they likely share the components of core biochemical machinery for DNA replication. Thus, underreplication provides a facile readout for late replication initiation and delayed fork progression.

The *Suppressor of UnderReplication* (*SuUR*) gene is essential for polytene chromosome under-replication in intercalary and pericentric heterochromatin (*Belyaeva et al., 1998*). In *SuUR* mutants, the DNA copy number in underreplicated regions is partially restored to almost reach those for fully polyploidized regions of the genome. *SuUR* encodes a protein (SUUR) containing a helicase domain

with homology to that of the SNF2/SWI2 family. The occupancy of ORC in intercalary and pericentric heterochromatin is not increased in *SuUR* mutants (*Sher et al., 2012*), and, thus, the increased replication of underreplicated regions is likely not due to the firing of additional origins. Rather, SUUR negatively regulates the rate of replication fork progression (*Nordman et al., 2014*) by an unknown mechanism. It has been proposed (*Posukh et al., 2015*) that retardation of the replisome by SUUR takes place via simultaneous physical association with the components of the fork (e.g., CDC45 and PCNA) (*Kolesnikova et al., 2013*; *Nordman et al., 2014*) and repressive chromatin proteins, such as HP1a (*Pindyurin et al., 2008*).

Using a newly developed proteomics approach, we discovered that SUUR forms a stable stoichiometric complex with a chromatin boundary protein Mod(Mdg4)-67.2. We demonstrate that SUUR and Mod(Mdg4)-67.2 together are required for maximal underreplication of intercalary heterochromatin and full activity of the *gypsy* insulator, thereby implicating insulators in obstructing replisome progression and the control of late DNA replication.

## Results

### Identification of SUMM4, the native form of SUUR in *Drosophila* embryos

To determine how SUUR functions in replication control, we sought to identify its native complex. Previous attempts to characterize the native form of SUUR by co-IP or tag-affinity purification gave rise to multiple putative binding partners (*Kolesnikova et al., 2013*; *Munden et al., 2018*; *Nordman et al., 2014*; *Pindyurin et al., 2008*). However, evaluating whether any of these proteins are present in a native SUUR complex is problematic because of the low abundance of SUUR, which also precludes its purification by conventional chromatography. Therefore, we developed a novel biochemical approach using embryonic extracts (which can be obtained in large quantities) that relies on partial purification by multistep FPLC (fast protein liquid chromatography) (*Figure 1A*) and shotgun proteomics of chromatographic fractions by quantitative LCMS. We term this technology MERCI for MS-Enabled Rapid protein Complex Identification ('Materials and methods').

Shotgun quantification of complex mixtures of polypeptides by LCMS is performed in two steps. First, the composition of the mixture is examined by information-dependent acquisitions (IDA) that establish protein identities based on MS1 and MS2 spectra of detected tryptic peptides. This information is used to compile a so-called 'ion library' (IL), which is then utilized to quantify spectral information obtained from the same samples by unbiased, data-independent acquisitions (DIA), sometimes termed sequential window acquisitions of all theoretical mass spectra (SWATH-MS/SWATH). Importantly, the depth of proteomic quantification is limited by the range of peptides in the IL originally built by IDA.

SUUR-specific peptides could not be found in ILs obtained from acquisitions of crude nuclear extracts or any fractions from the first, phosphocellulose, step (IL1, *Figure 1B*, *Supplementary file 1*), and therefore, SUUR could not be quantified in SWATH acquisitions of phosphocellulose fractions when IL1 alone is used as a reference. Thus, to measure the relative abundance of SUUR in phosphocellulose fractions, we augmented IL1 with the IL obtained by IDA of recombinant SUUR (ILR, *Figure 1B and C*). In ion libraries from subsequent chromatographic steps (IL2–IL5), peptides derived from native SUUR were detected (*Figure 1B*, *Supplementary file 1*) and used for quantification of cognate DIA/SWATH acquisitions (*Figure 1D–H*).

The final aspect of the MERCI algorithm calls for re-quantification of FPLC fraction SWATH acquisitions with an IL from the last step (IL5) that is enriched for peptides derived from SUUR and co-purifying polypeptides (*Figure 1A*) and includes only 140 proteins (*Figure 1B*, *Supplementary file 1*). In this fashion, scarce polypeptides (including SUUR and, potentially, SUUR-binding partners) that may not be detectable in earlier steps will not evade quantification. Purification profiles of proteins quantified in all five FPLC steps (132) were then artificially stitched into 83-point arrays of Z-scores (*Figure 1I*, *Supplementary file 2*). These profiles were Pearson-correlated with that of SUUR and ranked down from the highest Pearson coefficient, PCC (*Figure 2A*). Whereas the PCC numbers for the bottom 130 proteins lay on a smooth curve, the top two proteins, SUUR (PCC = 1.000) and Mod(Mdg4) (PCC = 0.939) fell above the extrapolated (by polynomial regression) curve (*Figure 2B*). Consistently, SUUR and Mod(Mdg4) exhibited nearly identical purification profiles in all five FPLC steps (*Figure 2C*), unlike

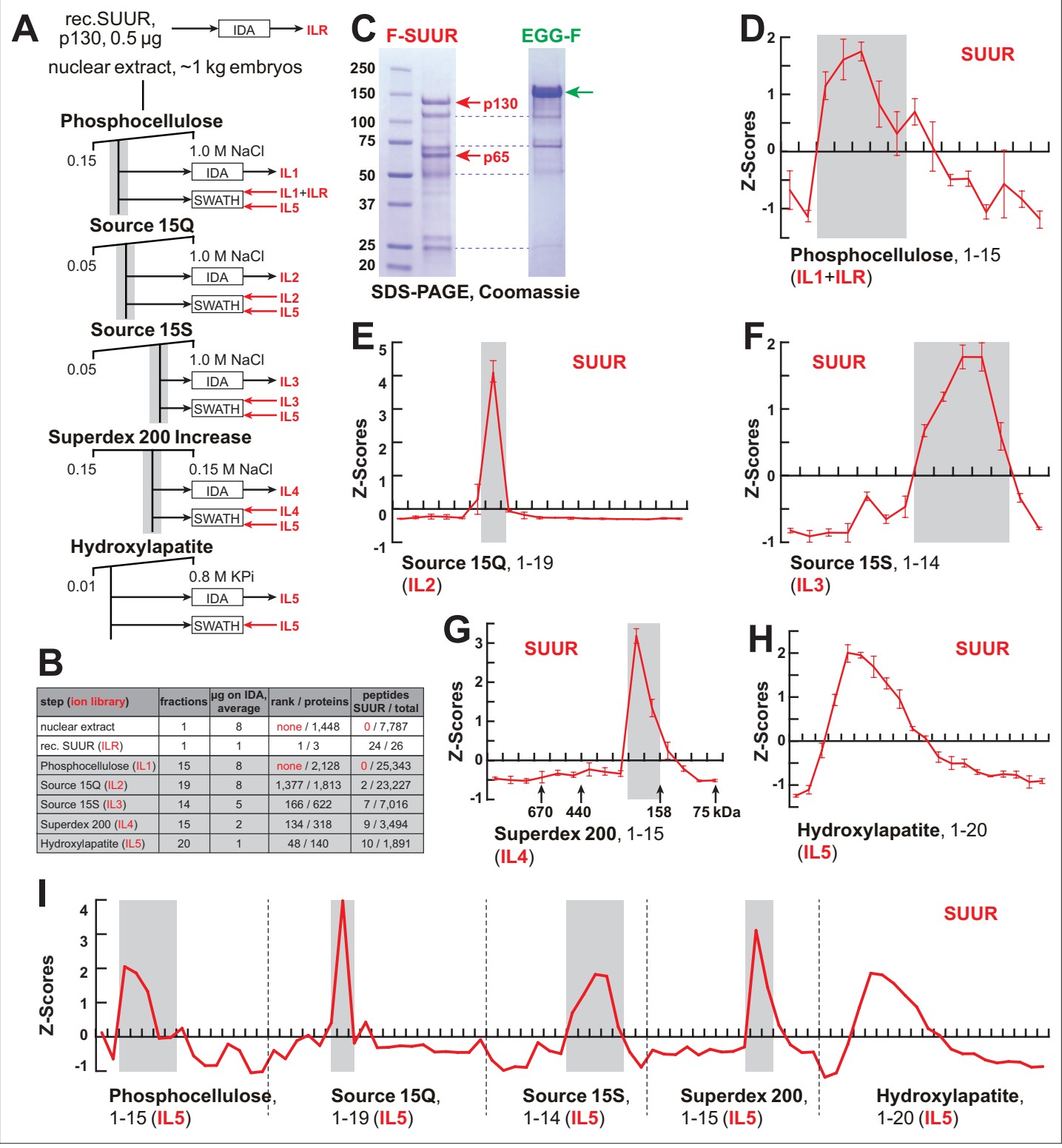

**Figure 1.** FPLC fractionation and MS-Enabled Rapid protein Complex Identification (MERCI) quantification of native SUUR. (**A**) Schematic of FPLC purification of the native form of SUUR using MERCI approach. ILR, ion library obtained by information-dependent acquisitions (IDA) of recombinant FLAG-SUUR; IL1-5, ion libraries obtained by IDA of FPLC fractions from chromatographic steps 1–5. KPi, potassium phosphate, pH 7.6. (**B**) Representation of SUUR in ion libraries ILR and IL1-5 (***Supplementary file 1***). Total number of identified proteins and the confidence rank of SUUR among them as well as the total number of detected peptides (95% confidence) and the number of SUUR-specific peptides are shown. (**C**) Recombinant FLAG-SUUR expressed in Sf9 cells. Identities of eight most prominent bands were determined by mass-spectroscopy. p130 and p65 correspond to full-

*Figure 1 continued on next page*

*Figure 1 continued*

length and C-terminally truncated FLAG-SUUR, respectively (red arrows). Other bands represent common Sf9-specific contaminants purified by FLAG chromatography (blue dashed lines), *cf.* purified EGG-F (green arrow). Molecular mass marker bands are indicated (kDa). (**D–H**) SWATH quantitation profiles of SUUR fractionation across individual FPLC steps. Ion libraries (IL) used for SWATH quantitation are shown at the bottom of each panel. Z-scores across indicated column fractions are plotted; error bars, standard deviations (N = 3). Gray rectangles, fraction ranges used for the next FPLC step; in (**G**), black arrows, expected peaks of globular proteins with indicated molecular masses in kDa. (**I**) SWATH quantitation profiles of SUUR fractionation across five FPLC steps. IL5 ion library was used for SWATH quantification.

The online version of this article includes the following source data and figure supplement(s) for figure 1:

**Source data 1.** FPLC column parameters (*Figure 1A*).

**Source data 2.** Recombinant proteins expressed in Sf9 cells and purified by FLAG affinitychromatography.

**Figure supplement 1.** Quantification of SUUR in chromatographic fractions.

the next two top-scoring proteins, EGG (PCC = 0.881) and CG6700 (PCC = 0.874) (*Figure 2—figure supplement 1A and B*). Also, HP1a (PCC = 0.503), which had been proposed to form a complex with SUUR (*Pindyurin et al., 2008*) did not co-purify with SUUR in any FPLC steps (*Figure 2—figure supplement 1C*).

Mod(Mdg4) is a BTB/POZ domain protein that functions as an adapter for architectural proteins that promote various aspects of genome organization (*Georgiev and Gerasimova, 1989*; *Gerasimova et al., 1995*). It is expressed as 26 distinct polypeptides generated by splicing in trans of a common 5′-end precursor RNA with 26 unique 3′-end precursors (*Büchner et al., 2000*). IL5 contained seven peptides derived from Mod(Mdg4) (99% confidence). Whereas four of them mapped to the common N-terminal 402 residues, three were specific to the C-terminus of a particular form, Mod(Mdg4)-67.2 (*Figure 2—figure supplement 2*). Peptides specific to other splice forms were not detected. We raised an antibody to the C-terminus of Mod(Mdg4)-67.2, designated ModT antibody, and analyzed size-exclusion column fractions by immunoblotting. Consistent with SWATH analyses (*Figures 1G and 2C*), SUUR and Mod(Mdg4)-67.2 polypeptides copurified as a complex with an apparent molecular mass of ~250 kDa (*Figure 2D*). Finally, we confirmed that SUUR specifically co-immunoprecipitated with Mod(Mdg4)-67.2 from embryonic nuclear extracts (*Figure 2E*). As a control, XNP co-immunoprecipitated with HP1a as shown previously (*Emelyanov et al., 2010*), but did not with SUUR or Mod(Mdg4) (*Figure 2E*). We conclude that SUUR and Mod(Mdg4) form a stable stoichiometric complex that we term SUMM4 (*S*uppressor of *U*nderreplication – *M*odifier of *Mdg4*).

## Biochemical activities of recombinant SUMM4 in vitro

We reconstituted recombinant SUMM4 complex by co-expressing FLAG-SUUR with Mod(Mdg4)-67.2-His$_6$ in Sf9 cells and purified it by FLAG affinity chromatography (*Figure 3A*). Mod(Mdg4)-67.2 is the predominant form of Mod(Mdg4) expressed in embryos (e.g., *Figure 2E*, left panel). Thus, minor Mod(Mdg4) forms may have failed to be identified by IDA in IL5 (*Figure 2—figure supplement 2A*). We discovered that FLAG-SUUR did not co-purify with another splice form, Mod(Mdg4)-59.1 (*Figure 3A*, *Figure 2—figure supplement 2C*). Whereas the identity of an ~100 kDa Mod(Mdg4)-67.2-His$_6$ band co-purifying with FLAG-SUUR was confirmed by mass-spec sequencing, the FLAG-purified material from Sf9 cells expressing FLAG-SUUR and Mod(Mdg4)-59.1 did not contain Mod(Mdg4)-specific peptides. Therefore, the shared N-terminus of Mod(Mdg4) (1–402) is not sufficient for interactions with SUUR. However, this result does not exclude a possibility that SUUR may form complex(es) with some of the other, low-abundance 24 splice forms of Mod(Mdg4). The SUUR-Mod(Mdg4)-67.2 interaction is specific as the second-best candidate from our correlation analyses (*Drosophila* SetDB1 ortholog EGG; *Figure 2B*) did not form a complex with FLAG-SUUR (*Figure 3—figure supplement 1A*), although it is associated with its known partner WDE, an ortholog of hATF7IP/mAM (*Wang et al., 2003*).

The N-terminus of SUUR contains a region homologous with SNF2-like DEAD/H helicase domains. Although SUUR requires its N-terminal domain to function in vivo (*Munden et al., 2018*), it has been hypothesized to be inactive as an ATPase (*Nordman and Orr-Weaver, 2015*). We analyzed the ability of recombinant SUUR and SUMM4 (*Figure 3A*) to hydrolyze ATP in vitro in comparison to recombinant *Drosophila* ISWI (*Figure 3—figure supplement 1B*). Purified recombinant Mod(Mdg4)-67.2 (*Figure 3A*) and a variant SUUR protein with a point mutation in the putative Walker A motif (K59A)

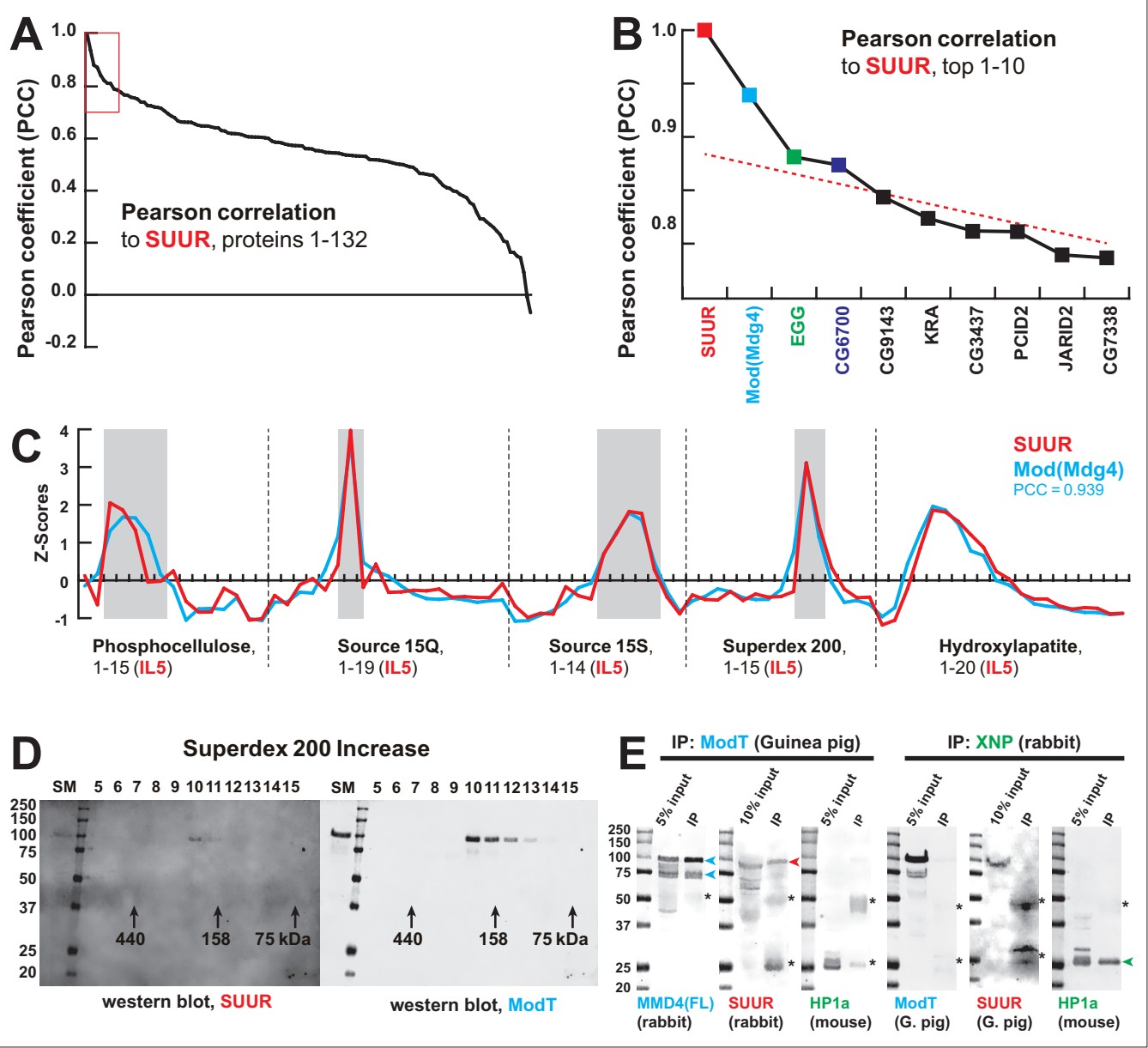

**Figure 2.** Identification of the SUMM4 complex by MS-Enabled Rapid protein Complex Identification (MERCI). (**A**) Pearson correlation of fractionation profiles for individual 132 proteins to that of SUUR, sorted from largest to smallest. Red box, the graph portion shown in (**B**). (**B**) Top 10 candidate proteins with the highest Pearson correlation to SUUR. Red dashed line, trend line extrapolated by polynomial regression (n = 5) from the bottom 130 proteins. (**C**) SWATH quantitation profiles of SUUR (red) and Mod(Mdg4) (cyan) fractionation across five FPLC steps, *Figure 1I*. IL5 ion library was used for SWATH quantification. (**D**) Western blot analyses of Superdex 200 fractions with SUUR and ModT antibodies, *Figure 1G*. Molecular mass markers are shown on the left (kDa). (**E**) Co-IP experiments. SUUR (red arrowhead) co-purifies from nuclear extracts with Mod(Mdg4)-67.2 (cyan arrowheads) but not HP1a (green arrowhead). Anti-XNP co-IPs HP1a but not SUUR of Mod(Mdg4)-67.2. Asterisks, IgG heavy and light chains detected due to antibody cross-reactivity. Mod(Mdg4)-67.2(FL) antibody recognizes all splice forms of Mod(Mdg4).

The online version of this article includes the following source data and figure supplement(s) for figure 2:

**Source data 1.** Western blots of chromatographic fractions.

**Source data 2.** Co-IP of SUMM4 subunits.

**Figure supplement 1.** Comparisons of SWATH quantification profiles for protein fractionation.

**Figure supplement 2.** Identification of Mod(Mdg4)-67.2 as a subunit of the SUMM4 complex.

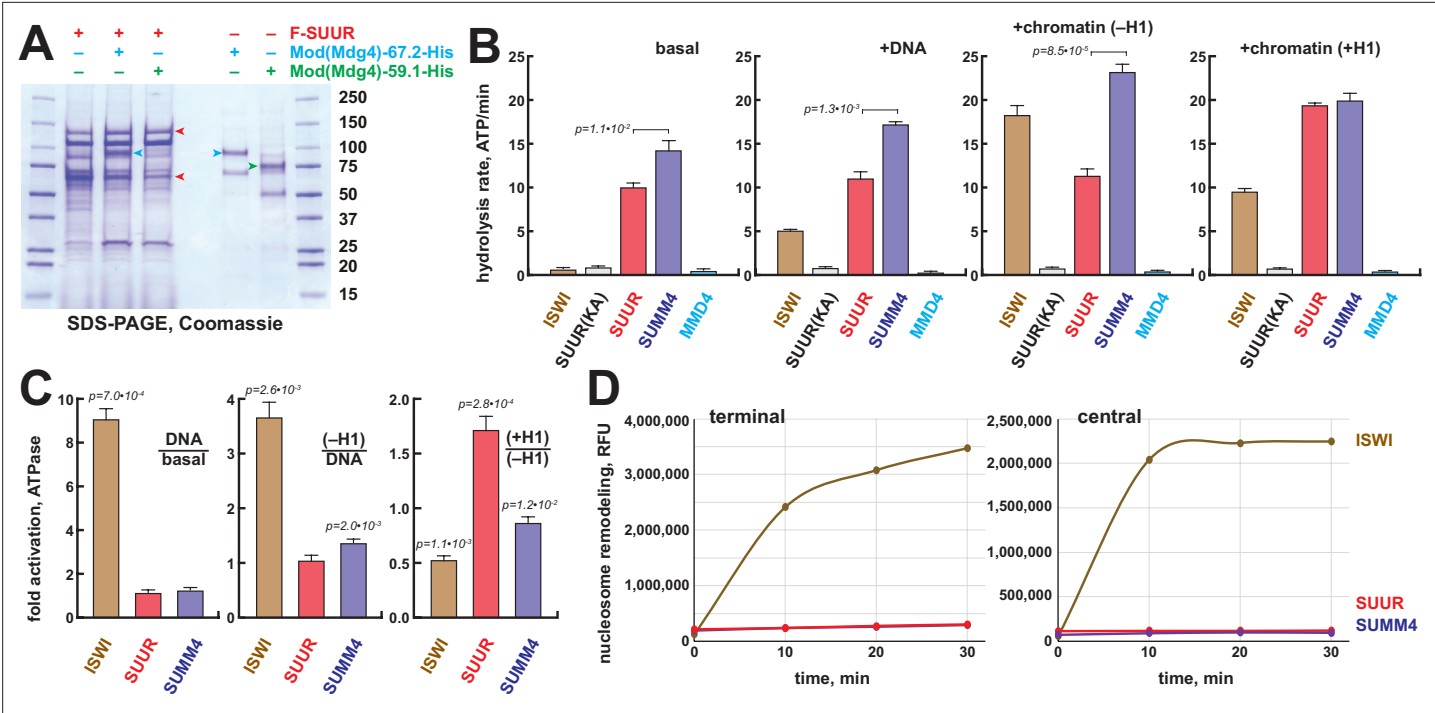

**Figure 3.** Biochemical activities of recombinant SUMM4. (**A**) Recombinant SUMM4. Mod(Mdg4)-His$_6$, 67.2 (p100, cyan arrowhead) and 59.1 (p75, green arrowhead) splice forms were co-expressed with FLAG-SUUR (red arrowheads, p130 and p65) or separately in Sf9 cells and purified by FLAG or Ni-NTA affinity chromatography. Mod(Mdg4)-67.2 forms a specific complex with SUUR. Identities of the 130, 100, 75, and 65 kDa protein bands from FLAG- and Ni-NTA-purified material were determined by mass spectroscopy. (**B**) ATPase activities of recombinant ISWI (brown bars), FLAG-SUUR (red bars), and SUMM4 (FLAG-SUUR + Mod(Mdg4)-67.2-His$_6$, purple bars). Equimolar amounts of proteins were analyzed in reactions in the absence or presence of plasmid DNA or equivalent amounts of reconstituted oligonucleosomes,±H1. SUUR(KA) and MMD4, ATPases activities of K59A mutant of SUUR (gray bars) and Mod(Mdg4)-67.2-His$_6$ (cyan bars). Hydrolysis rates were converted to moles ATP per mole protein per minute. All reactions were performed in triplicate (N=3), error bars represent standard deviations. p-Values for statistically significant differences are indicated (Mann–Whitney test). (**C**) DNA- and nucleosome-dependent stimulation or inhibition of ATPase activity. The activities were analyzed as in (**B**). Statistically significant differences are shown (Mann–Whitney test). (**D**) Nucleosome sliding activities by EpiDyne-PicoGreen assay (see 'Materials and methods') with 5 nM of recombinant ISWI, SUUR, or SUMM4. Reaction time courses are shown for terminally (6-N-66) and centrally (50-N-66) positioned mononucleosomes (**Figure 3—figure supplement 2B–E**). RFU, relative fluorescence units produced by PicoGreen fluorescence.

The online version of this article includes the following source data and figure supplement(s) for figure 3:

**Source data 1.** Recombinant proteins expressed in Sf9 cells and purified by FLAG or Ni-NTA affinity chromatography.

**Figure supplement 1.** Recombinant proteins and biochemical substrates.

**Figure supplement 1—source data 1.** Recombinant proteins expressed in Sf9 cells and purified by FLAG or Ni-NTA affinity chromatography.

**Figure supplement 1—source data 2.** Recombinant proteins expressed in Sf9 or *E. coli* cells and purified by FLAG, Ni-NTA, or chitin affinity chromatography.

**Figure supplement 1—source data 3.** SDS-PAGE of salt dialyzed chromatin ±H1.

**Figure supplement 1—source data 4.** Micrococcal nuclease (MNase) digest of salt-dialyzed chromatin ± H1, 1.25% agarose gel, ethidium-stained.

**Figure supplement 1—source data 5.** Micrococcal nuclease (MNase) digest of salt-dialyzed chromatin ± H1, 3% agarose gel, ethidium-stained (chromatosome stop assay).

**Figure supplement 2.** EpiDyne-PicoGreen biochemical assay.

were used as negative controls (**Figure 3A**, **Figure 3—figure supplement 1B**). Contrary to the prediction, both SUUR and SUMM4 exhibited strong ATPase activities (**Figure 3B**). SUMM4 was 1.4- to 2-fold more active than SUUR alone, indicating that Mod(Mdg4)-67.2 stimulates SUUR enzymatic activity. We then examined whether DNA and nucleosomes can stimulate the activity of SUUR. To this end, we reconstituted oligonucleosomes on plasmid DNA (**Figure 3—figure supplement 1C–E**). Linker histone H1-containing chromatin was also used as a substrate/cofactor because SUUR has been demonstrated to physically interact with H1 (**Andreyeva et al., 2017**). In contrast to ISWI, SUUR was not stimulated by addition of DNA or nucleosomes and moderately (by about 70%) activated by

H1-containing oligonucleosomes (*Figure 3C*) consistent with its reported direct physical interaction with H1 (*Andreyeva et al., 2017*).

We examined the nucleosome remodeling activities of SUUR and SUMM4; specifically, their ability to expose a positioned DNA motif in the EpiDyne-PicoGreen assay ('Materials and methods' and *Figure 3—figure supplement 2A*). Centrally or terminally positioned mononucleosomes were efficiently mobilized by ISWI and human BRG1 in a concentration- and time-dependent manner (*Figure 3—figure supplement 2B–E*). In contrast, SUUR and SUMM4 did not reposition either nucleosome (*Figure 3D*). Thus, SUUR and SUMM4 do not possess a detectable remodeling activity and may resemble certain other SNF2-like enzymes (e.g., RAD54) that utilize the energy of ATP hydrolysis to mediate alternate DNA translocation reactions (*Jaskelioff et al., 2003*).

## The distribution of SUMM4 complex in vivo

We examined the positions of SUUR and Mod(Mdg4)-67.2 within polytene chromosomes by indirect immunofluorescence (IF) and discovered that they overlap at numerous locations (*Figure 4A*, *Figure 4—figure supplement 1A and B*). In late endo-S phase, when SUUR exhibited a characteristic distribution, it co-localized with Mod(Mdg4)-67.2 at numerous (hundreds of) loci along the chromosome arms (*Figure 4—figure supplement 1B*). Mod(Mdg4)-67.2 was present at classical regions of SUUR enrichment, such as underreplicated domains in 75C and 89E (*Figure 4—figure supplement 1A*). The chromocenter, which consists of underreplicated pericentric heterochromatin, contains SUUR but did not show occupancy by Mod(Mdg4)-67.2 (*Figure 4—figure supplement 1A*). Conversely, there were multiple sites of Mod(Mdg4)-67.2 localization that were free of SUUR (*Figure 4—figure supplement 1A and B*). Individual pixel intensities of IF signals for SUUR and Mod(Mdg4)-67.2 were plotted as a 2D scatter plot (*Figure 4—figure supplement 1C*) and were found to exhibit a weak positive correlation ($R^2 = 0.278$). Consistent with the possible multi-phasic relative distribution of SUUR and Mod(Mdg4)-67.2 (*Figure 4—figure supplement 1B*), the 2D plot encompassed four distinct areas, where SUUR and Mod(Mdg4)–67.2-were co-localized, enriched separately, or absent (*Figure 4—figure supplement 1D*). When regions of SUUR-alone and Mod(mdg4)-67.2-alone enrichment were excluded, and only the regions of their apparent colocalization were considered, the anti-SUUR and anti-ModT signals exhibited a strong positive correlation ($R^2 = 0.568$, *Figure 4—figure supplement 1D*).

The existence of chromosome loci heavily enriched for Mod(Mdg4)-67.2 but devoid of SUUR suggests that there are additional native form(s) of Mod(Mdg4)-67.2, either as an individual polypeptide or in complex(es) other than SUMM4. When we fractionated *Drosophila* nuclear extract using a different progression of FPLC steps (*Figure 4—figure supplement 2A*), we found that Mod(Mdg4)-67.2 can form a megadalton-sized complex that did not contain SUUR (*Figure 4—figure supplement 2B–D*). Therefore, a more intricate pattern of Mod(Mdg4)-67.2 distribution likely reflects loading of both SUMM4 and an alternative Mod(Mdg4)-67.2-containing complex.

We tested whether SUUR and Mod(Mdg4) loading into polytene chromosomes were mutually dependent using mutant alleles of *SuUR* and *mod(mdg4)*. *SuUR*[ES] is a null allele of *SuUR* (*Makunin et al., 2002*). *mod(mdg4)*[m9] is a null allele with a deficiency that removes gene regions of the shared 5′-end precursor and eight specific 3′-precursors (*Savitsky et al., 2016*). *mod(mdg4)*[u1] contains an insertion of a *Stalker* element in the last coding exon of Mod(Mdg4)-67.2 3′-precursor (*Gerasimova et al., 1995*), and thus is predicted only to disrupt expression of this isoform. *SuUR*[ES] and *mod(mdg4)*[u1] are homozygous viable, and *mod(mdg4)*[m9] is recessive adult pharate lethal. Although homozygous *mod(mdg4)*[m9] animals die after the pupal stage, they survive until late third-instar larvae (L3). Therefore, this allele cannot be used to study adult phenotypes, but it is possible to analyze its effects in L3, such as on polytene chromosome structure. Importantly, however, since the homozygous progeny is produced by heterozygous parents, the recessive phenotypes would not reveal themselves until the maternally loaded protein and RNA are exhausted (diluted and/or degraded) by late larval stages, as frequently occurs for other *Drosophila* mutants.

We could not detect Mod(Mdg4)-67.2 expression in homozygous *mod(mdg4)*[m9] L3 salivary glands by immunoblotting, whereas *mod(mdg4)*[u1] expressed a truncated polypeptide (cf., ~70 kDa and ~100 kDa, *Figure 4—figure supplement 3A*). The truncated 70 kDa polypeptide failed to load into polytene chromosomes (*Figure 4B*, *Figure 4—figure supplement 3B*). As shown previously, SUUR could not be detected in *SuUR*[ES] chromosomes. Since homozygous *mod(mdg4)*[m9] L3 larvae

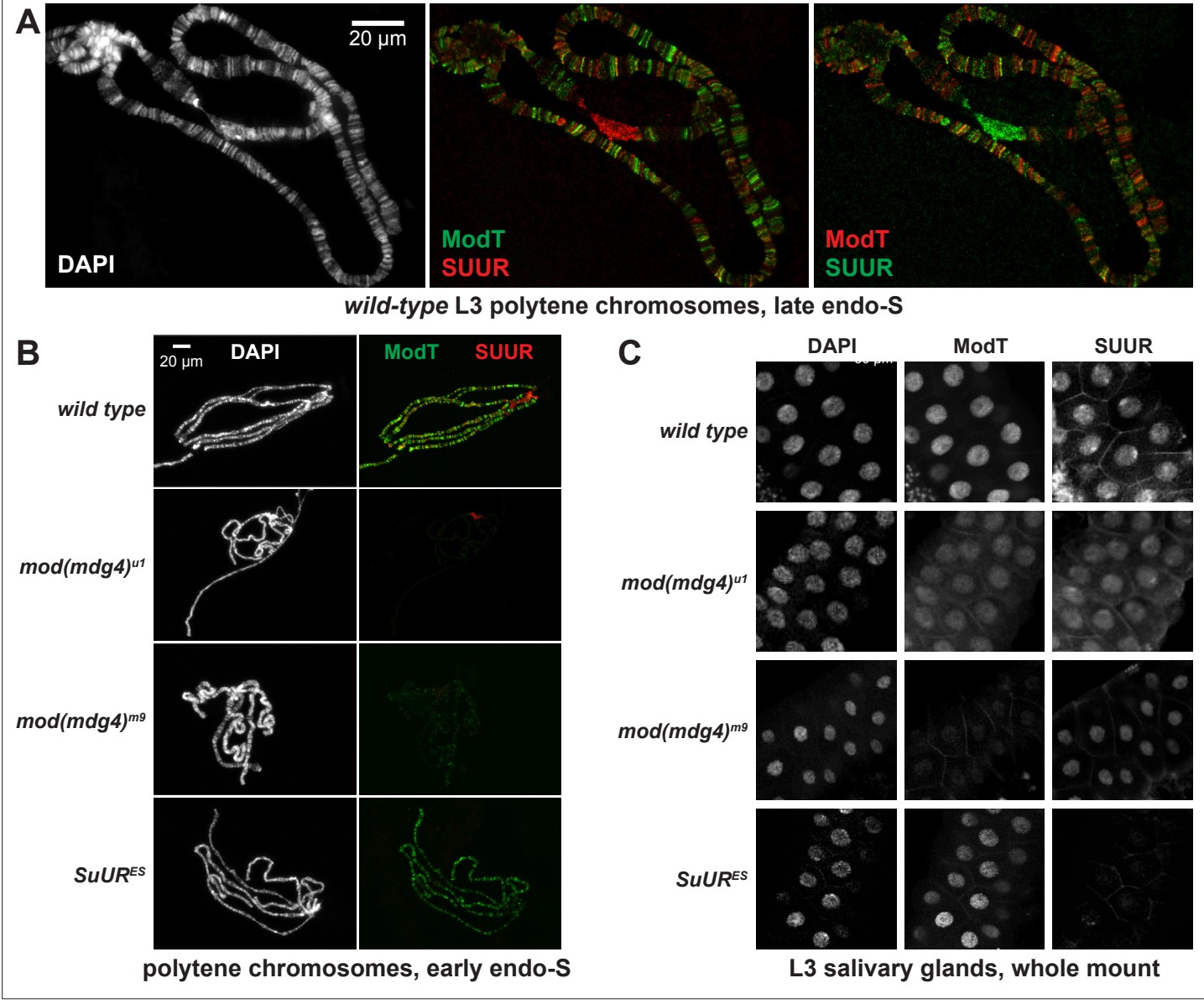

**Figure 4.** Spatiotemporal distribution of SUMM4 in vivo. (**A**) Colocalization of SUUR and Mod(Mdg4)-67.2 in *wild-type* polytene chromosomes. Localization patterns of Mod(Mdg4)-67.2 and SUUR in L3 polytene chromosomes were analyzed by indirect immunofluorescence (IF) staining. The polytene spread fragment (3L and 3R arms) corresponds to a nucleus in late endo-S phase, according to PCNA staining (*Figure 4—figure supplement 1A*). Left panel, DAPI staining shows the overall chromosome morphology. Middle panel, ModT (green) and SUUR (red) signals overlap extensively in euchromatic arms. Right panel, a colocalization image with swapped red (ModT) and green (SUUR) channels is shown for comparison. Note the additional strong ModT IF loci that are SUUR-free as well as Mod(Mdg4)-67.2-free SUUR in pericentric 3LR. (**B**) SUUR loading into chromosomes during early endo-S phase is compromised in *mod(mdg4)* mutants. *SuUR* mutation does not appreciably change the distribution of Mod(Mdg4)-67.2. Endo-S timing was established by PCNA staining (*Figure 4—figure supplement 3B*). (**C**) Abnormal subcellular distribution of SUMM4 subunits in *mod(mdg4)* and *SuUR* mutants. L3 salivary glands were fixed and whole-mount-stained with DAPI, ModT, and SUUR antibodies. Whereas both polypeptides are mostly nuclear in wild-type, they are partially mis-localized to the cytoplasm in *mod(mdg4)ᵘ¹* mutant.

The online version of this article includes the following source data and figure supplement(s) for figure 4:

**Figure supplement 1.** Spatial distribution of SUUR and Mod(Mdg4)-67.2 in polytene chromosomes and analyses of their colocalization.

**Figure supplement 2.** Alternative complex(es) of Mod(Mdg4)-67.2.

**Figure supplement 2—source data 1.** FPLC column parameters (*Figure 4—figure supplement 2A*).

**Figure supplement 2—source data 2.** Western blots of chromatographic fractions.

**Figure supplement 2—source data 3.** Western blots of chromatographic fractions.

*Figure 4 continued on next page*

*Figure 4 continued*

**Figure supplement 2—source data 4.** Western blots of chromatographic fractions.

**Figure supplement 3.** Spatiotemporal distribution of SUMM4 subunits in polytene chromosomes of *mod(mdg4)* and *SuUR* mutant alleles.

**Figure supplement 3—source data 1.** Western blots of salivary gland lysates.

were produced by *inter se* crosses of heterozygous parents, the very low amounts of Mod(Mdg4)-67.2 in *mod(mdg4)$^{m9}$* polytene chromosomes (barely above the detection limit) were presumably maternally contributed.

The absence (or drastic decrease) of Mod(Mdg4)-67.2 also strongly reduced the loading of SUUR (*Figure 4B*, *Figure 4—figure supplement 3B*). The normal distribution pattern of SUUR in polytene chromosomes is highly dynamic (*Andreyeva et al., 2017*; *Kolesnikova et al., 2013*). SUUR is initially loaded in chromosomes at the onset of endo-S phase and then redistributes through very late endo-S, when it accumulates in underreplicated domains and pericentric heterochromatin. In both *mod(mdg4)* mutants, we observed a striking absence of SUUR in euchromatic arms of polytene chromosomes during early endo-S (*Figure 4B*, *Figure 4—figure supplement 3B*), which indicates that the initial deposition of SUUR is dependent on its interactions with Mod(Mdg4). Although SUUR deposition slightly recovered by late endo-S, it was still several fold weaker than that in wild-type control. Potentially, in the absence of Mod(Mdg4), SUUR may be tethered to intercalary and pericentric heterochromatin loci by direct binding with linker histone H1 as shown previously (*Andreyeva et al., 2017*). Finally, the gross subcellular distribution of SUUR also strongly correlated with that of Mod(Mdg4): a mis-localization of truncated Mod(Mdg4)-67.2 from nuclear to partially cytoplasmic was accompanied by a similar mis-localization of SUUR (*Figure 4C*). This result indicates that the truncation of Mod(Mdg4) in *mod(mdg4)$^{u1}$* may have an antimorphic effect by mis-localization and deficient chromatin loading of interacting polypeptides, including SUUR (*Figure 4C*) and others (*Figure 4—figure supplement 2B–D*).

## The role of SUMM4 as an effector of the insulator/chromatin barrier function

Mod(Mdg4)-67.2 does not directly bind DNA but instead is tethered by a physical association with zinc finger factor Suppressor of Hairy Wing, Su(Hw) (*Gause et al., 2001*). Su(Hw) directly binds to consensus sequences that are present in *gypsy* transposable elements and are also widely distributed across the *Drosophila* genome in thousands of copies (*Adryan et al., 2007*). Mod(Mdg4)-67.2 was previously shown to be essential for the insulator activity of *gypsy* (*Gerasimova et al., 1995*), which functions in vivo to prevent enhancer–promoter interactions and establish a barrier to the propagation of chromatin forms (*Cai and Levine, 1995*; *Roseman et al., 1993*). We therefore tested whether SUMM4 contributes to the *gypsy* insulator functions.

The *ct$^6$* allele of *Drosophila* contains a *gypsy* element inserted between the wing enhancer and promoter of the gene *cut*. The insertion inactivates *cut* expression and results in abnormal wing development (*Figure 5A*). We discovered that both *mod(mdg4)$^{u1}$* and *SuUR$^{ES}$* mutations partially suppressed this phenotype (*Figure 5A*) and significantly increased the wing size compared to *ct$^6$* allele alone (*Figure 5B*). Thus, both subunits of SUMM4 are required to mediate the full enhancer-blocking activity of *gypsy*. Interestingly, the double, *SuUR$^{ES}$* and *mod(mdg4)$^{u1}$*, mutant produced an additional suppression of the *ct$^6$* phenotype compared to that by *mod(mdg4)$^{u1}$* alone (*Figure 5A*, red arrowhead), which suggests that SUUR may contribute to the insulator function in the absence of Mod(Mdg4)-67.2.

Another insulator assay makes use of a collection of *P{SUPor-P}* insertions that contain the *white* reporter flanked by 12 copies of *gypsy* Su(Hw)-binding sites (*Figure 5C*, top). When *P{SUPor-P}* is inserted in heterochromatin, *white* is protected from silencing, resulting in red eyes (*Roseman et al., 1995*). Both *mod(mdg4)$^{u1}$* and *SuUR$^{ES}$* relieved the chromatin barrier function of Su(Hw) sites, causing repression of *white* (*Figure 5C*). We conclude that SUMM4 is an insulator complex that contributes to the enhancer-blocking and chromatin boundary functions of *gypsy* by a mechanism schematized in *Figure 6A and B*.

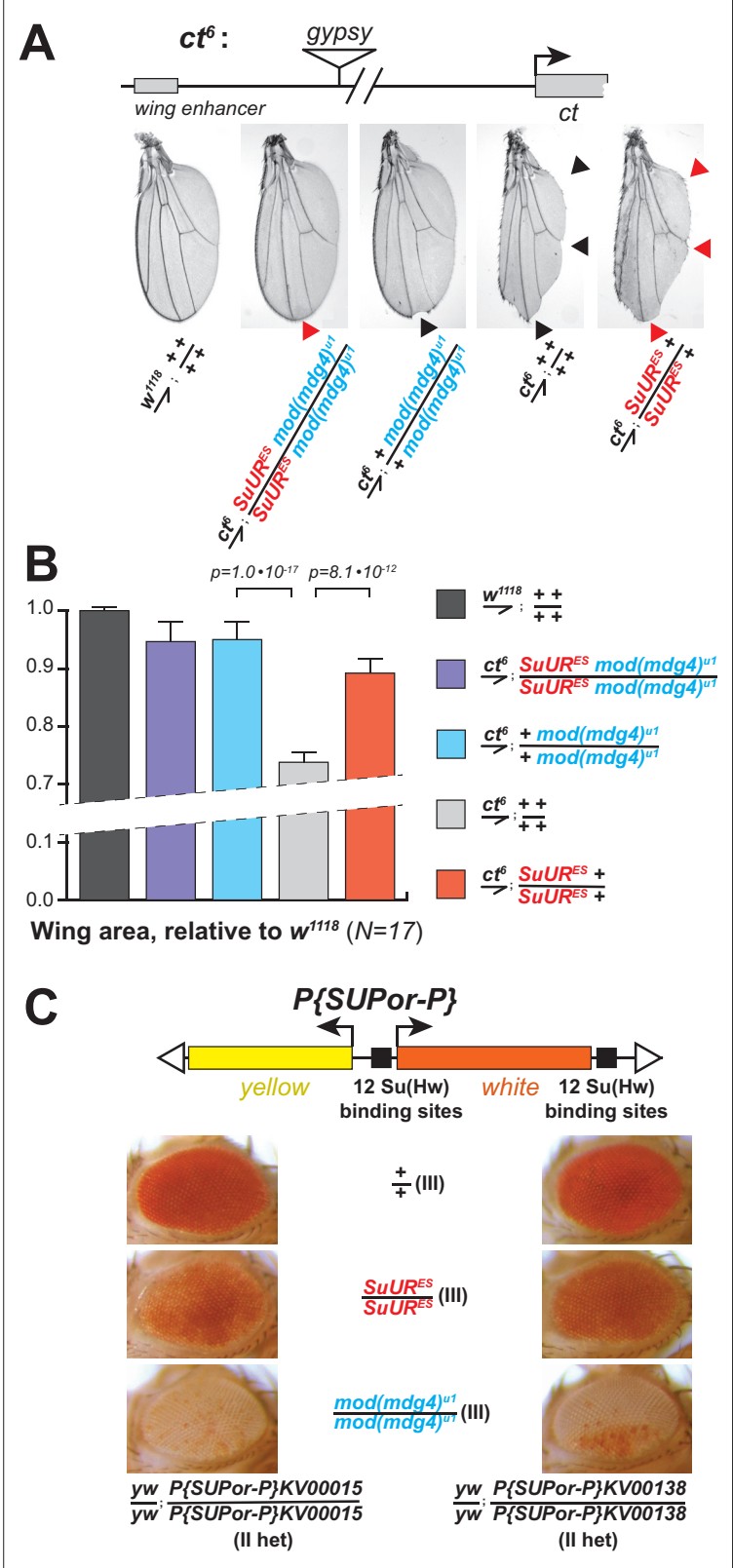

**Figure 5.** Biological functions of SUMM4 in the regulation of gene expression. (**A**) SUMM4 subunits are required for the enhancer-blocking activity in $ct^6$. Top: schematic diagram of the $ct^6$ reporter system; the *gypsy* retrotransposon is inserted in between the wing enhancer and promoter of *cut* (**Bag et al., 2019**). Bottom left: the appearance of wild-type adult wing; bottom right: the appearance of $ct^6$ adult wing in the wild-type background.

*Figure 5 continued on next page*

*Figure 5 continued*

*SuUR^ES* and *mod(mdg4)^u1* alleles are recessive suppressors of the *ct^6* phenotype. Red and black arrowheads point to distinct anatomical features of the wing upon *SuUR* mutation. (**B**) Relative sizes (areas) of wings in adult male flies of the indicated phenotypes were measured (N=17) as described in 'Materials and methods.' p-Values for statistically significant differences are indicated (t-test). (**C**) SUMM4 subunits are required for the chromatin barrier activity of Su(Hw) binding sites. Top: schematic diagram of the *P{SUPor-P}* reporter system (*Bellen et al., 2004*); clustered 12 copies of *gypsy* Su(Hw) binding sites flanks the transcription unit of *white*. KV00015 and KV00138 are *P{SUPor-P}* insertions in pericentric heterochromatin of 2L. *SuUR^ES* and *mod(mdg4)^u1* alleles are recessive suppressors of the boundary that insulates *white* from heterochromatin encroachment.

## The role of SUMM4 in the regulation of DNA replication in polytene chromosomes

A similar, chromatin partitioning-related mechanism may direct the function of SUUR in the establishment of underreplication in late-replicating intercalary heterochromatin domains of polytene chromosomes (*Figure 6C*). It has been long known that 3D chromosome partitioning maps show an 'uncanny alignment' with replication timing maps (*Rhind and Gilbert, 2013*). To examine the possible roles of SUMM4 in underreplication, we measured DNA copy number genome-wide in salivary glands of L3 larvae by next-generation sequencing (NGS). In *w^1118* control salivary glands, the DNA copy profile revealed large (>100 kbp) domains of reduced ploidy (*Figure 7A*), similar to previous reports (*Andreyeva et al., 2017*; *Sher et al., 2012*; *Yarosh and Spradling, 2014*). Excluding pericentric and subtelomeric heterochromatin, we called 70 underreplicated regions (*Table 1*) in euchromatic arms, as described in 'Materials and methods'.

In both *SuUR* and *mod(mdg4)^m9* null larvae, we observed statistically significant suppression of underreplication in intercalary heterochromatin (*Figure 7B*, *Figure 7—figure supplement 1A*, *Table 1*). In line with its lack of accumulation within the chromocenter of polytene chromosomes (*Figure 4A*), Mod(Mdg4) was largely dispensable for underreplication in pericentric heterochromatin. The NGS data strongly correlated with qPCR measurements of DNA copy numbers (*Figure 7C and D*). Furthermore, cytological evidence in the 75C region supported the molecular analyses in that both mutants exhibited a brighter DAPI staining of the 75C1-2 band than that in *w^1118*, indicative of higher DNA content (*Figure 7D*). Importantly, consistent with the role of Mod(Mdg4)-dependent insulators in the establishment of underreplication, the boundaries of underreplicated domains frequently encompass multiple clustered Su(Hw) binding sites (*Figure 7C and D*).

Uniformly, *SuUR* mutation gave rise to a stronger relief of underreplication than that produced by the *mod(mdg4)^m9* null allele (*Table 1*). This result can be explained by embryonic deposition of functional Mod(Mdg4) proteins and RNA by heterozygous mothers, unlike the complete absence of SUUR throughout the life cycle of the homozygous viable and fertile *SuUR^ES* animals. Although third-instar larvae are >1000-fold larger, volume-wise, than the embryos, persistent Mod(Mdg4)-67.2 can still be detected in polytene chromosomes of these larvae by IF despite its dilution and degradation

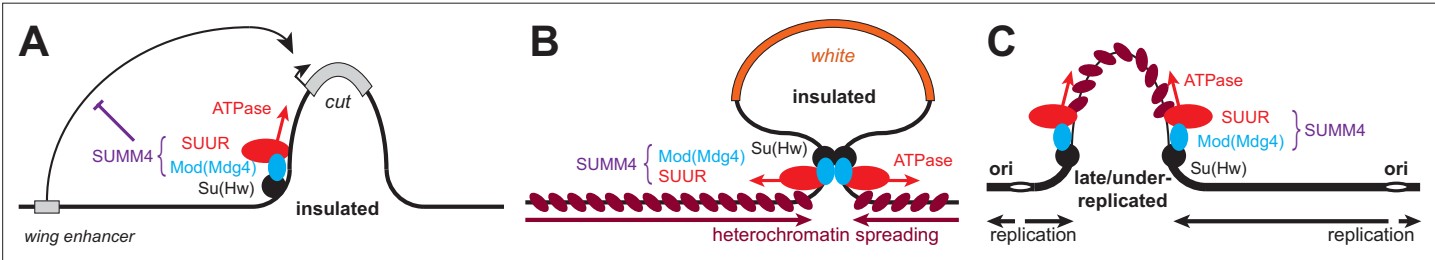

**Figure 6.** Schematic models for the biological functions of SUMM4 in the regulation of gene expression and DNA replication. (**A**) Schematic model for the function of SUMM4 in blocking enhancer–promoter interactions in the *ct^6* locus. A *gypsy* mobile element inserted between wing enhancer and gene *cut* encompasses multiple Su(Hw) binding sites. (**B**) Schematic model for the function of SUMM4 in establishing a chromatin barrier in heterochromatin-inserted *P{SUPor-P}* elements. The reporter gene *white* is flanked on both sides by 12 copies of *gypsy* insulator element. (**C**) Schematic model for a putative function of SUMM4 in blocking/retardation of replication fork progression in intercalary heterochromatin domains. Black oval, Su(Hw) protein bound to a *gypsy* insulator element(s); cyan oval, Mod(Mdg4)-67.2 protein tethered to Su(Hw); red oval, SUUR protein associated with Mod(Mdg4)-67.2 in SUMM4 complex; brown ovals represent heterochromatin components; gray rectangles, gene *cut* and its upstream wing enhancer; orange rectangle, gene *white*.

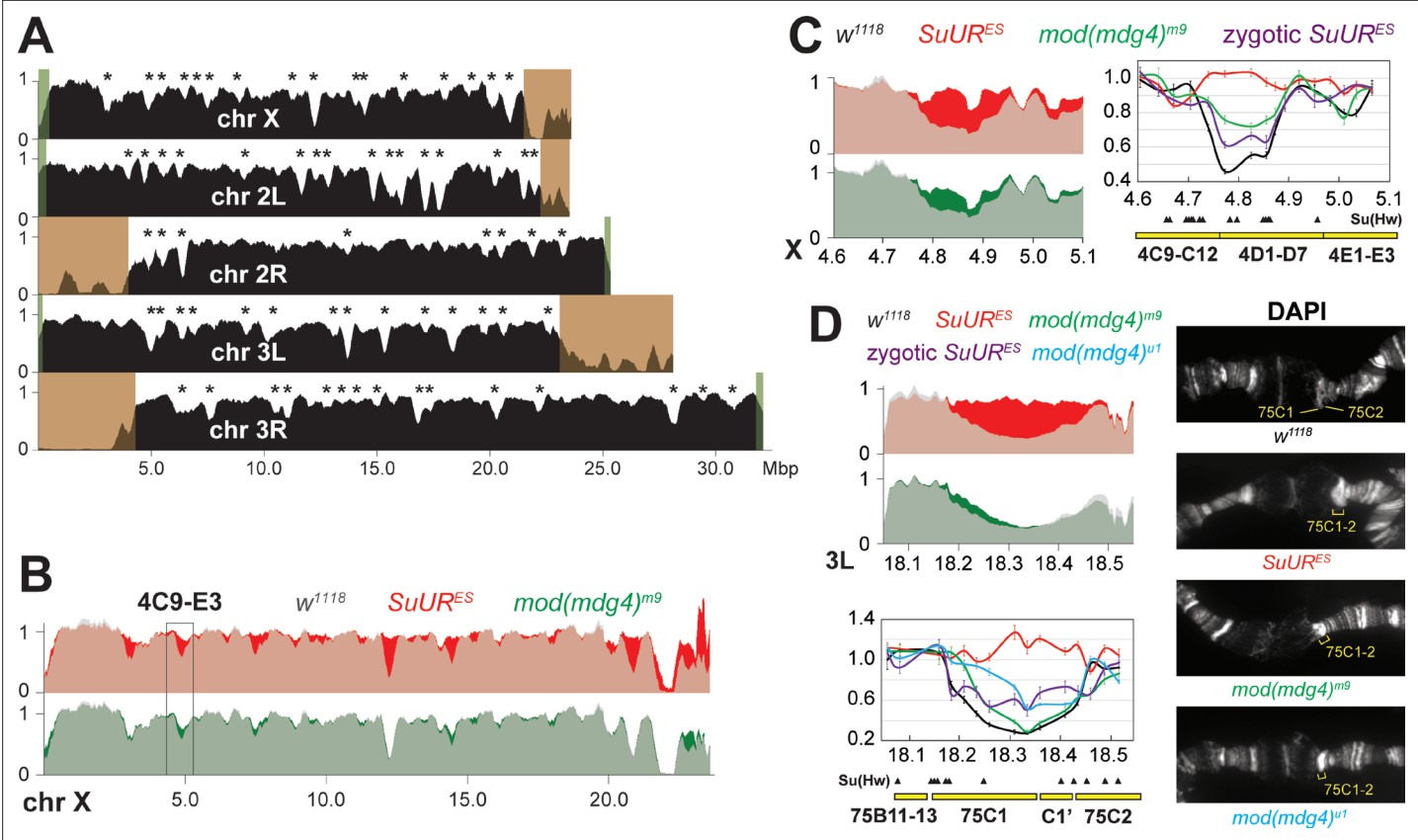

**Figure 7.** Biological functions of SUMM4 in the regulation of DNA replication. (**A**) Genome-wide analyses of DNA copy numbers in *Drosophila* salivary gland cells (*w1118* control). DNA from L3 salivary glands was subjected to high-throughput sequencing. DNA copy numbers (normalized to diploid embryonic DNA) are shown for chromosomes X, II, and III. Chromosome arms are indicated in white. Brown- and green-shades boxes, mapped pericentric and telomeric heterochromatin regions (*Hoskins et al., 2015*), respectively. Asterisks, positions of underreplicated domains (*Table 1*). Genomic coordinates in Megabase pairs are indicated at the bottom. (**B**) Analyses of DNA copy numbers in *Drosophila* salivary gland cells from wild-type and mutant alleles. Normalized DNA copy numbers are shown across the X chromosome. The control trace (*w1118* allele) is shown as semitransparent light gray in the foreground; *SuURES* (homozygous null) and *mod(mdg4)m9* (zygotic null from crosses of heterozygous parents) traces are shown in the background in red and green, respectively; their overlaps with *w1118* traces appear as lighter shades of colors. Black box, 4C9-E3 cytological region. (**C**) Close-up view of DNA copy numbers in region 4C9-E3 from high-throughput sequencing data are presented as in (**B**). DNA copy numbers were also measured independently by real-time qPCR. The numbers were calculated relative to embryonic DNA and normalized to a control intergenic region. The X-axis shows chromosome positions (in Megabase pairs) of target amplicons. Black, *w1118*; red, *SuURES* (homozygous null); green, *mod(mdg4)m9* (zygotic null from crosses of heterozygous parents); purple, *SuURES* (zygotic null from crosses of heterozygous parents). Error bars represent the confidence interval (N=9, see 'Materials and methods'). Black arrowheads, positions of mapped Su(Hw) binding sites (*Nègre et al., 2010*). Yellow boxes show approximate boundaries of cytogenetic bands. (**D**) Close-up view of DNA copy numbers by high-throughput sequencing and by qPCR for region 75B11-C2 and DAPI-stained polytene chromosome segments around cytological regions 75B-75C. Yellow lines or brackets in DAPI images indicate positions of 75C1 and 75C2 bands (*w1118* control) or fused 75C1-2 band (mutants); cyan, *mod(mdg4)u1* (homozygous null); for other designations see (**C**).

The online version of this article includes the following source data and figure supplement(s) for figure 7:

**Source data 1.** Primer sequences used for qPCR.

**Figure supplement 1.** Biological functions of SUMM4 in the regulation of underreplication.

(*Figure 4B*, *Figure 4—figure supplement 3B*). In contrast, unlike L3, first-instar larvae (L1) are nearly identical in size to the embryos. Therefore, since the endoreplication cycles initiate in embryos and L1, in *mod(mdg4)m9* animals the first few out of 10–11 rounds of chromosome polytenization take place with an almost normal amount of Mod(Mdg4) present, which may substantially limit the effect of *mod(mdg4)m9* mutation on underreplication as measured in L3.

Seemingly, there is a contradiction between a strong effect that *mod(mdg4)* null mutation has on the loading of SUUR in polytene chromosomes (*Figure 4B*) and a weaker effect on underreplication

**Table 1.** Underreplicated domains and suppression of underreplication in (UR) SUMM4 subunit mutant alleles.
Domains of UR in euchromatic arms of polytene chromosomes were called in $w^{1118}$ as described in 'Materials and methods.' Their genomic coordinates, approximate cytological location ('Cyto band'), and average DNA copy numbers ('<CN>') in homozygous $w^{1118}$, $SuUR^{ES}$, and $mod(mdg4)^{m9}$ L3 larvae are shown. <CN> numbers were normalized to the average DNA copy numbers across euchromatic genome. UR percent recovery levels were calculated as (<CN> mut $-$<CN>$_{w1118}$) / (1 $-$ <CN> $_{w1118}$); negative numbers indicate increased UR. UR p-values were calculated using the DESeq2 package by averaging the Wald test p-values of each 5 kbp bin significantly different than the $w^{1118}$ signal. UR was called as suppressible by a mutant if p<0.01; p-values for regions that exhibit a statistically significant recovery of UR are shown in bold blue. Averages of <CN> across all called underreplicated domains and averages of percent Recovery across all suppressible underreplicated domains ('<Recovery>', bottom row) were adjusted for each underreplicated domain length; calculation errors = standard deviations.

| | Chromosome coordinates | | | | | UR, $w^{1118}$ | UR, $SuUR^{ES}$ | | | UR, $mod(mdg4)^{m9}$ | | |
|---|---|---|---|---|---|---|---|---|---|---|---|---|
| N | Arm | Left | Right | Cyto band | Length | <CN> | <CN> | Recovery (%) | p-Value | <CN> | Recovery (%) | p-Value |
| 1 | X | 2,950,001 | 3,140,000 | 3C3-C7 | 190,000 | 0.51 | 0.93 | 86 | **7.3E-05** | 0.58 | 14 | 1.1E-02 |
| 2 | X | 4,710,001 | 4,900,000 | 4C15-D5 | 190,000 | 0.56 | 0.96 | 92 | **3.9E-04** | 0.81 | 57 | **6.9E-05** |
| 3 | X | 4,965,001 | 5,070,000 | 4E1-E2 | 105,000 | 0.72 | 0.86 | 50 | **5.6E-04** | 0.80 | 28 | 1.4E-02 |
| 4 | X | 6,415,001 | 6,525,000 | 6A1-B1 | 110,000 | 0.71 | 0.90 | 65 | **1.4E-03** | 0.80 | 29 | **7.3E-03** |
| 5 | X | 7,335,001 | 7,560,000 | 7B1-B4 | 225,000 | 0.65 | 0.98 | 95 | **1.2E-03** | 0.79 | 40 | **2.8E-03** |
| 6 | X | 7,750,001 | 7,865,000 | 7B7-C1 | 115,000 | 0.64 | 0.94 | 84 | **3.0E-09** | 0.84 | 55 | **5.2E-07** |
| 7 | X | 8,880,001 | 9,005,000 | 8B5-C2 | 125,000 | 0.73 | 0.86 | 50 | **5.5E-03** | 0.76 | 9 | **4.6E-03** |
| 8 | X | 9,405,001 | 9,555,000 | 8D12-E7 | 150,000 | 0.72 | 0.91 | 67 | **3.6E-04** | 0.85 | 47 | **3.6E-03** |
| 9 | X | 11,170,001 | 11,325,000 | 10A10-B3 | 155,000 | 0.67 | 0.84 | 53 | **3.2E-03** | 0.78 | 35 | **2.6E-03** |
| 10 | X | 12,040,001 | 12,430,000 | 11A2-A10 | 390,000 | 0.38 | 0.97 | 94 | **1.4E-08** | 0.42 | 6 | **6.8E-03** |
| 11 | X | 13,950,001 | 14,100,000 | 12D1-E1 | 150,000 | 0.69 | 0.72 | 10 | 1.0E-02 | 0.73 | 14 | 1.4E-02 |
| 12 | X | 14,290,001 | 14,565,000 | 12E7-F1 | 275,000 | 0.51 | 0.94 | 87 | **4.1E-04** | 0.69 | 36 | **8.1E-04** |
| 13 | X | 17,925,001 | 18,030,000 | 16F3-F5 | 105,000 | 0.67 | 0.99 | 98 | **1.7E-15** | 0.90 | 68 | **3.4E-05** |
| 14 | X | 20,000,001 | 20,105,000 | 19A4-B1 | 105,000 | 0.79 | 1.12 | 157 | **1.4E-13** | 0.82 | 12 | **6.1E-03** |
| 15 | X | 20,525,001 | 21,020,000 | 19D2-E7 | 495,000 | 0.50 | 0.97 | 93 | **1.3E-07** | 0.51 | 2 | **4.9E-03** |
| 16 | X | 21,630,001 | 22,450,000 | 20A5-C1 | 820,000 | 0.04 | 0.32 | 29 | **1.8E-03** | 0.06 | 2 | **6.4E-03** |
| 17 | X | 22,550,001 | 22,995,000 | 20C2-F3 | 445,000 | 0.48 | 0.81 | 64 | **7.8E-05** | 0.74 | 51 | **3.5E-04** |
| 18 | 2L | 3,920,001 | 4,025,000 | 24D1-D4 | 105,000 | 0.63 | 0.93 | 81 | **7.9E-07** | 0.80 | 46 | **5.9E-05** |
| 19 | 2L | 4,585,001 | 4,790,000 | 25A2-A5 | 205,000 | 0.66 | 0.99 | 98 | **1.9E-08** | 0.78 | 36 | **1.3E-03** |
| 20 | 2L | 5,400,001 | 5,510,000 | 25E1-E4 | 110,000 | 0.82 | 0.99 | 95 | **4.0E-08** | 0.90 | 45 | **8.3E-03** |
| 21 | 2L | 6,155,001 | 6,320,000 | 26B9-C2 | 165,000 | 0.74 | 1.08 | 130 | **7.3E-14** | 0.88 | 54 | **4.7E-04** |
| 22 | 2L | 9,030,001 | 9,150,000 | 29F8-30A2 | 120,000 | 0.76 | 0.98 | 93 | **1.5E-04** | 0.95 | 79 | **3.3E-03** |
| 23 | 2L | 11,535,001 | 11,795,000 | 32F2-33A1 | 260,000 | 0.44 | 0.90 | 83 | **2.9E-04** | 0.57 | 24 | **1.5E-03** |
| 24 | 2L | 12,215,001 | 12,340,000 | 33D3-E1 | 125,000 | 0.58 | 0.86 | 66 | **3.6E-11** | 0.75 | 40 | **1.1E-04** |
| 25 | 2L | 12,765,001 | 12,970,000 | 33F5-34A3 | 205,000 | 0.55 | 0.91 | 79 | **8.8E-04** | 0.73 | 40 | **7.0E-05** |
| 26 | 2L | 14,685,001 | 15,010,000 | 35B4-B8 | 325,000 | 0.41 | 0.88 | 80 | **5.7E-04** | 0.54 | 23 | **7.2E-04** |
| 27 | 2L | 15,295,001 | 15,735,000 | 35D1-D4 | 440,000 | 0.49 | 0.76 | 53 | **2.3E-05** | 0.54 | 9 | **4.0E-03** |
| 28 | 2L | 15,770,001 | 15,900,000 | 35D4-D6 | 130,000 | 0.54 | 0.87 | 71 | **4.5E-08** | 0.68 | 31 | **6.7E-04** |
| 29 | 2L | 15,925,001 | 16,240,000 | 35D6-F1 | 315,000 | 0.29 | 0.90 | 87 | **6.7E-07** | 0.38 | 12 | **1.4E-05** |
| 30 | 2L | 16,925,001 | 17,375,000 | 36B4-C7 | 450,000 | 0.23 | 0.89 | 85 | **1.4E-04** | 0.26 | 4 | **4.3E-03** |
| 31 | 2L | 17,515,001 | 18,100,000 | 36C10-E4 | 585,000 | 0.34 | 0.87 | 80 | **5.0E-06** | 0.36 | 2 | **3.7E-03** |

*Table 1 continued on next page*

Table 1 continued

| N | Arm | Chromosome coordinates Left | Right | Cyto band | Length | UR, $w^{1118}$ <CN> | UR, $SuUR^{ES}$ <CN> | Recovery (%) | p-Value | UR, $mod(mdg4)^{m9}$ <CN> | Recovery (%) | p-Value |
|---|-----|------|-------|-----------|--------|------|------|--------------|---------|------|--------------|---------|
| 32 | 2L | 18,160,001 | 18,300,000 | 36E6-F2 | 140,000 | 0.67 | 0.99 | 97 | 3.3E-06 | 0.90 | 69 | 3.1E-06 |
| 33 | 2L | 20,110,001 | 20,290,000 | 38C1-C4 | 180,000 | 0.48 | 0.69 | 41 | 8.9E-04 | 0.46 | –5 | 1.8E-03 |
| 34 | 2L | 20,485,001 | 20,620,000 | 38C8-D1 | 135,000 | 0.77 | 0.98 | 93 | 1.0E-06 | 0.99 | 97 | 2.1E-05 |
| 35 | 2L | 21,400,001 | 21,550,000 | 39D3-E2 | 150,000 | 0.10 | 0.15 | 5 | 3.2E-03 | 0.14 | 3 | 4.4E-03 |
| 36 | 2L | 21,805,001 | 22,125,000 | 40A4-E4 | 320,000 | 0.53 | 0.94 | 87 | 6.9E-05 | 0.54 | 1 | 9.5E-03 |
| 37 | 2R | 4,875,001 | 5,050,000 | 41C4-D1 | 175,000 | 0.35 | 0.86 | 78 | 2.3E-10 | 0.34 | –1 | 4.0E-03 |
| 38 | 2R | 5,410,001 | 5,535,000 | 41F1-F3 | 125,000 | 0.58 | 0.79 | 50 | 1.1E-03 | 0.52 | –13 | 2.2E-03 |
| 39 | 2R | 6,290,001 | 6,505,000 | 42A14-B1 | 215,000 | 0.13 | 0.50 | 42 | 9.3E-04 | 0.14 | 1 | 2.7E-03 |
| 40 | 2R | 13,620,001 | 13,760,000 | 50B6-C3 | 140,000 | 0.63 | 0.95 | 88 | 4.1E-18 | 0.78 | 41 | 1.3E-05 |
| 41 | 2R | 20,355,001 | 20,540,000 | 56F17-57A5 | 185,000 | 0.56 | 0.92 | 83 | 2.0E-06 | 0.71 | 35 | 8.2E-04 |
| 42 | 2R | 21,830,001 | 21,945,000 | 58A2-A4 | 115,000 | 0.72 | 0.95 | 83 | 1.1E-05 | 0.71 | –3 | 2.2E-02 |
| 43 | 2R | 23,145,001 | 23,320,000 | 59D1-D6 | 175,000 | 0.62 | 1.04 | 110 | 1.3E-22 | 0.67 | 13 | 7.7E-03 |
| 44 | 3L | 4,840,001 | 5,100,000 | 64C1-C5 | 260,000 | 0.38 | 0.92 | 87 | 3.5E-08 | 0.40 | 3 | 6.6E-03 |
| 45 | 3L | 5,385,001 | 5,510,000 | 64C15-D3 | 125,000 | 0.51 | 0.88 | 76 | 1.9E-22 | 0.73 | 45 | 6.0E-09 |
| 46 | 3L | 6,290,001 | 6,485,000 | 65A11-B3 | 195,000 | 0.52 | 0.89 | 77 | 4.9E-05 | 0.71 | 38 | 1.2E-04 |
| 47 | 3L | 9,180,001 | 9,300,000 | 67A1-A7 | 120,000 | 0.67 | 0.97 | 90 | 6.5E-09 | 0.73 | 20 | 1.0E-02 |
| 48 | 3L | 10,000,001 | 10,195,000 | 67D3-D10 | 195,000 | 0.62 | 0.97 | 93 | 4.4E-13 | 0.79 | 44 | 5.7E-06 |
| 49 | 3L | 13,085,001 | 13,220,000 | 70A1-A2 | 135,000 | 0.66 | 1.01 | 104 | 3.6E-09 | 0.89 | 66 | 2.9E-06 |
| 50 | 3L | 13,550,001 | 13,855,000 | 70B6-C4 | 305,000 | 0.26 | 0.95 | 94 | 1.8E-06 | 0.39 | 18 | 7.3E-04 |
| 51 | 3L | 15,175,001 | 15,500,000 | 71B7-D3 | 325,000 | 0.39 | 0.94 | 89 | 5.6E-04 | 0.46 | 10 | 3.7E-03 |
| 52 | 3L | 17,115,001 | 17,240,000 | 73F1-74A1 | 125,000 | 0.71 | 1.02 | 106 | 4.3E-05 | 0.84 | 45 | 2.7E-03 |
| 53 | 3L | 18,175,001 | 18,525,000 | 75B11-75D2 | 350,000 | 0.45 | 0.87 | 76 | 6.8E-05 | 0.47 | 4 | 4.6E-03 |
| 54 | 3L | 20,555,001 | 20,695,000 | 77D1-77E3 | 140,000 | 0.60 | 1.02 | 106 | 2.2E-22 | 0.84 | 61 | 3.6E-11 |
| 55 | 3R | 6,060,001 | 6,310,000 | 83D2-E4 | 250,000 | 0.70 | 0.92 | 72 | 7.6E-04 | 0.63 | –22 | 1.0E-02 |
| 56 | 3R | 6,495,001 | 6,635,000 | 83F1-84A1 | 140,000 | 0.53 | 0.96 | 91 | 7.8E-08 | 0.71 | 39 | 2.2E-04 |
| 57 | 3R | 6,915,001 | 7,055,000 | 84B1-B2 | 140,000 | 0.64 | 0.93 | 80 | 3.9E-04 | 0.82 | 49 | 1.9E-05 |
| 58 | 3R | 7,550,001 | 7,785,000 | 84D9-84E2 | 235,000 | 0.44 | 0.80 | 65 | 8.0E-06 | 0.51 | 12 | 4.2E-03 |
| 59 | 3R | 10,450,001 | 10,660,000 | 86B6-C4 | 210,000 | 0.55 | 0.98 | 97 | 8.1E-11 | 0.66 | 25 | 7.6E-04 |
| 60 | 3R | 10,910,001 | 11,140,000 | 88C15-86D4 | 230,000 | 0.45 | 0.94 | 89 | 2.3E-10 | 0.46 | 2 | 2.3E-03 |
| 61 | 3R | 12,050,001 | 12,165,000 | 87A5-B1 | 115,000 | 0.63 | 0.96 | 88 | 9.9E-24 | 0.81 | 49 | 5.9E-09 |
| 62 | 3R | 12,745,001 | 12,935,000 | 87C8-D4 | 190,000 | 0.67 | 0.89 | 68 | 7.5E-05 | 0.60 | –21 | 1.1E-02 |
| 63 | 3R | 14,935,001 | 15,055,000 | 88D8-D10 | 120,000 | 0.70 | 0.88 | 61 | 7.6E-06 | 0.84 | 47 | 1.0E-04 |
| 64 | 3R | 16,670,001 | 16,970,000 | 89D6-E5 | 300,000 | 0.40 | 0.92 | 87 | 2.7E-09 | 0.47 | 10 | 3.2E-03 |
| 65 | 3R | 17,160,001 | 17,355,000 | 89F1-90A2 | 195,000 | 0.62 | 0.94 | 84 | 1.0E-03 | 0.86 | 64 | 2.8E-04 |
| 66 | 3R | 20,085,001 | 20,290,000 | 92C4-E1 | 205,000 | 0.61 | 0.81 | 53 | 1.5E-03 | 0.71 | 26 | 3.6E-03 |
| 67 | 3R | 20,340,001 | 20,525,000 | 92E4-E12 | 185,000 | 0.58 | 0.96 | 91 | 5.0E-05 | 0.79 | 50 | 7.2E-04 |
| 68 | 3R | 22,110,001 | 22,295,000 | 94A2-A4 | 185,000 | 0.61 | 0.93 | 83 | 3.4E-11 | 0.76 | 39 | 3.0E-04 |
| 69 | 3R | 28,005,001 | 28,295,000 | 98B7-C3 | 290,000 | 0.40 | 0.91 | 85 | 2.5E-05 | 0.60 | 32 | 6.9E-04 |

*Table 1 continued on next page*

*Table 1 continued*

| | Chromosome coordinates | | | | | UR, $w^{1118}$ | UR, $SuUR^{ES}$ | | | UR, $mod(mdg4)^{m9}$ | | |
|---|---|---|---|---|---|---|---|---|---|---|---|---|
| N | Arm | Left | Right | Cyto band | Length | <CN> | <CN> | Recovery (%) | p-Value | <CN> | Recovery (%) | p-Value |
| 70 | 3R | 28,370,001 | 28,480,000 | 98C5-D2 | 110,000 | 0.73 | 0.98 | 94 | 1.2E-09 | 0.91 | 66 | 4.3E-07 |

| | | | |
|---|---|---|---|
| UR domains: 70<br><Length> : 216 ± 64 kbp<br>Average <CN> across all UR domains: 0.49 ± 0.08 | | Suppressed UR domains: 69<br><Length> : 217 ± 64 kbp<br><Recovery> : 78 ± 11% | Suppressed UR domains: 60<br><Length> : 225 ± 67 kbp<br><Recovery> : 26 ± 9% |

(*Figure 7B–D*, *Figure 7—figure supplement 1A and B*, *Table 1*). However, the SUUR occupancy is examined in L3 after the maternal *mod(mdg4)* product is nearly eliminated (*Figure 4B*). On the other hand, the DNA copy number, although also measured in L3 (*Figure 7B–D*, *Figure 7—figure supplement 1A and B*, *Table 1*), is a product of multiple rounds of endoreplication that initiate before Mod(Mdg4) is exhausted. To validate the putative effect of maternally contributed SUMM4 on the establishment of underreplication, we performed qPCR measurements of DNA copy numbers in salivary glands of homozygous *SuUR* animals produced by *inter se* crosses of heterozygous $SuUR^{ES}$/+ parents (*Figure 7C and D*, zygotic $SuUR^{ES}$). Similar to the maternal Mod(Mdg4), the initial maternal contribution of SUUR partially limited the reversal of underreplication in cytological regions 4D and 75C. Thus, when the *SuUR* and *mod(mdg4)* null mutant animals are similarly derived from heterozygous mothers that deposit wild-type gene product into their progeny, the mutant underreplication phenotypes in the third-instar larval salivary gland are essentially indistinguishable. Finally, we analyzed the effect of homozygous $mod(mdg4)^{u1}$ mutation, which is viable and fertile, on DNA copy numbers in the 75C underreplicated domain by qPCR and cytologically (*Figure 7D*). We observed a substantially stronger suppression of underreplication than that in $mod(mdg4)^{m9}$, presumably due to the absence of maternal contribution of full-length Mod(Mdg4)-67.2.

We conclude that SUUR and Mod(Mdg4)-67.2 act together as subunits of stable SUMM4 complex, which is required for the establishment of underreplication in the intercalary heterochromatin domains of *Drosophila* polytene chromosome.

## Discussion

### MERCI is a powerful new approach to characterize stable stoichiometric protein complexes

We present here a facile method, termed MERCI, to rapidly identify subunits of stable native complexes by only partial chromatographic purification. It allows one to circumvent the conventional, rate-limiting approach to purify proteins to apparent homogeneity. Since a multistep FPLC scheme invariably leads to an exponential loss of material, reducing the number of purification steps in the MERCI protocol allows identification of rare complexes, such as SUMM4, which may be present in trace amounts in native sources. On the other hand, MERCI obviates introduction of false-positives frequently associated with tag purification of ectopically expressed targets that render results less reliable. Notably, MERCI is not limited to analyses of known polypeptides since it is readily amenable to fractionation of native factors based on a correlation with their biochemical activities in vitro.

The dissection of protein interactome by extract fractionation on orthogonal FPLC columns and MS-based approaches has been previously attempted (*Havugimana et al., 2012*; *Shatsky et al., 2016*). However, unlike the newly developed MERCI approach, these studies were aimed at comprehensive, proteome-wide analyses, which managed to only yield data for the most abundant complexes. The major distinction of the MERCI protocol is that it is targeted toward a particular protein (SUUR in this study). The crucial final stage of the MERCI algorithm is re-quantification of all acquired SWATH data using a library acquired from fractions of the last column (IL5, *Figure 1A, B, and I*). The target protein and co-purifying polypeptides are substantially enriched after several chromatographic steps and, thus, yield a greater number of detected peptides, which helps a more precise quantification. Although SWATH allows reliable measurement of picogram amounts of proteins (*Figure 1—figure supplement 1A and B*), the range of quantified polypeptides is always limited by those present in IDA (ion libraries). For low-abundance proteins, such as SUUR and Mod(Mdg4), specific peptides

are not detectable by IDA in earlier chromatographic steps (*Supplementary file 1*). Consequently, SWATH quantification using only the cognate ion libraries would not discern the near perfect co-fractionation of SUUR and Mod(Mdg4) in all five steps (*Figure 2C*), precluding identification of the SUUR-Mod(Mdg4) complex (*Figure 2B and C*).

One limitation of the MERCI protocol is its failure to measure the absolute amounts of identified polypeptides. For instance, quantification of SWATH data (*Figure 1D–H*) measures the relative (to reference proteins and each other) amounts of SUUR across fractions. To measure the absolute levels of SUUR, a semi-quantitative approach was used by building a titration curve from SWATH acquisitions of known amounts of recombinant SUUR (*Figure 1—figure supplement 1A and B*). We estimated the amount of SUUR in the nuclear extract (~140 pg in 25 µg total protein, *Figure 1—figure supplement 1B*) and in individual fractions from all chromatographic steps (*Figure 1—figure supplement 1C*). Although in five FPLC steps we achieved >3000-fold purification of SUUR, it remained only ~2% pure (*Figure 1—figure supplement 1D*). A progressive loss of material precludes further purification (300 ng of SUUR in 16 µg total protein). Thus, the SUMM4 complex would be nearly impossible to purify to homogeneity from a substantial amount of starting material (~1 kg *Drosophila* embryos, ~2.5 g protein), suggesting that SUMM4 could not be identified by the classical FPLC approach.

## SUMM4 regulates the function of *gypsy* insulator elements

Both subunits of SUMM4 contribute to the known functions of *gypsy* insulator (*Figure 5A–C*). Although a *SuUR* mutation decreased the insulator activity, the suppression was universally weaker than that by *mod(mdg4)^u1^*. It is possible that SUUR is not absolutely required for the establishment of the insulator. For instance, the loss of SUMM4 may be compensated by the alternative complex of Mod(Mdg4)-67.2 (*Figure 4—figure supplement 2*). Furthermore, the *mod(mdg4)^u1^* allele is expected to have an antimorphic function since it can mis-localize interacting partner proteins, including SUUR itself (*Figure 4C*). Interestingly, *SuUR* has been previously characterized as a weak suppressor of variegation of the *white^m4h^* X chromosome inversion allele, which places the *white* gene near pericentric heterochromatin (*Belyaeva et al., 2003*). In contrast, *SuUR* mutation enhances variegation in the context of insulated, heterochromatin-positioned *white* (*Figure 5C*). Therefore, this phenotype is unrelated to the putative *Su(var)* function of *SuUR* but, rather, is insulator-dependent.

## ATP-dependent motor proteins are required for the establishment of chromatin barrier and chromosome partitioning

Our discovery and analyses of SUMM4 provide a biochemical link between ATP-dependent motor factors and the activity of insulators in the regulation of gene expression and chromatin partitioning. Insulator elements organize the genome into chromatin loops (*Gerasimova et al., 1995*) that are involved in the formation of topologically associating domains [TADs] (*Peterson et al., 2021*; *Rowley et al., 2017*; *Szabo et al., 2019*). In mammals, CTCF-dependent loop formation requires ATP-driven motor activity of SMC complex cohesin (*Davidson et al., 2019*). In contrast, CTCF and cohesin are thought to be dispensable for chromatin 3D partitioning in *Drosophila* (*Matthews and White, 2019*). Instead, the larger, transcriptionally inactive domains (canonical TADs) are interspersed with smaller active compartmental domains, which themselves represent TAD boundaries (*Rowley et al., 2017*). It has been proposed that in *Drosophila*, domain organization does not rely on architectural proteins but is established by transcription-dependent, A-A compartmental (gene-to-gene) interactions (*Rowley et al., 2017*). However, *Drosophila* TAD boundaries are enriched for architectural proteins other than CTCF (*Van Bortle et al., 2014*), and their roles have not been tested in loss-of-function models. Thus, it is possible that in *Drosophila*, instead of CTCF, the 3D partitioning of the genome is facilitated by another group of insulator proteins, such as Su(Hw) and SUMM4, that together associate with class 3 insulators (*Schwartz et al., 2012*).

Moreover, SUUR may provide the DNA motor function to promote a physical separation of active and inactive loci and help establish chromosome contact domains (*Figure 6A–C*). We propose that within the SUMM4 complex, SUUR utilizes its putative ATP-dependent motor activity to translocate along chromatin strands, thus facilitating the establishment of higher-order structures that isolate promoters from enhancers (*Figure 6A*) and stabilize DNA loops/domains to prevent unrestricted heterochromatin encroachment (*Figure 6B*) and penetration of replication forks (*Figure 6C*). The translocation model is consistent with observations of an asymmetric, selective occupancy of SUUR

away from its initial sites of deposition via Su(Hw)-Mod(Mdg4) binding toward inside of intercalary heterochromatin regions but not outside (*Figure 7—figure supplement 1C*; *Filion et al., 2010*), which may be facilitated by physical interactions between SUUR and linker histone H1 enriched in intercalary heterochromatin (*Andreyeva et al., 2017*). It has been reported that another *Drosophila* BTB/POZ domain insulator protein CP190 forms a complex with a DEAD-box helicase Rm62 that contributes to the insulator activity (*Lei and Corces, 2006*). Thus, ATP-dependent motor proteins may represent an obligatory component of the insulator complex machinery.

## SUMM4 mediates known biological functions of SUUR

Our discovery explains previous observations about biological functions of SUUR. For instance, the initial deposition of SUUR and its colocalization with PCNA has been proposed to depend on direct physical interaction with components of the replisome (*Kolesnikova et al., 2013*). Our model indicates that, instead, the apparent colocalization of SUUR with PCNA throughout endo-S phase (*Figure 4—figure supplement 3B*) may be caused by a replication fork retardation at insulator sites. SUUR is deposited in chromosomes as a subunit of SUMM4 complex at thousands of loci by tethering via Mod(Mdg4)-Su(Hw) interactions. As replication forks progress through the genome, they encounter insulator complexes where replication machinery pauses for various periods of time before resolving the obstacle. Thus, the increased co-residence time of PCNA and SUUR manifests cytologically as their partial colocalization. With the progression of endo-S phase, some of the SUMM4 insulator complexes are evicted and, thus, the number of SUUR-positive loci is decreased, until eventually the replication fork encounters nearly completely impenetrable insulators demarcating the underreplicated domain boundaries.

This mechanism is especially plausible given that boundaries of intercalary heterochromatin loci very frequently encompass multiple, densely clustered Su(Hw) binding sites (e.g., *Figure 7C and D*). We examined the data from genome-wide proteomic analyses for Su(Hw) and SUUR performed by DamID in Kc167 cells (*Filion et al., 2010*). Strikingly, Su(Hw) DamID-measured occupancy does not exhibit a discrete pattern expected of a DNA-binding factor. Instead, it appears broadly dispersed, together with SUUR, up to tens of kbp away from mapped Su(Hw) binding sites (*Figure 7—figure supplement 1C*). Interestingly, when hidden Markov modeling was applied to the DamID data, Su(Hw), Mod(Mdg4)-67.2, and SUUR occupancies were found to strongly correlate genome-wide in a novel chromatin form ('malachite') that frequently demarcates the boundaries of intercalary heterochromatin (*Khoroshko et al., 2016*). These observations strongly corroborate the translocation model for the mechanism of action of SUMM4. According to this model, upon tethering to DNA-bound Su(Hw), SUMM4 traverses the underreplicated region, which helps to separate it in a contact domain. As DNA within the underreplicated region is tracked by SUUR (*Figure 6C*), it is brought into a transient close proximity with both SUMM4 and the associated Su(Hw) protein, which is detected by DamID (or ChIP) as an expanded occupancy pattern.

The deceleration of SUUR-bound replication forks was also invoked as an explanation for the apparent role of SUUR in the establishment of epigenetic marking of intercalary heterochromatin (*Posukh et al., 2015*). We propose that global epigenetic modifications observed in the *SuUR* mutant likely do not directly arise from derepression of the replisome as suggested but, rather, result from the coordinate insulator-dependent regulatory functions of SUUR in both the establishment of a chromatin barrier and DNA replication control (*Figure 6B and C*).

## Architectural proteins can attenuate replication forks and regulate replication timing

Our work demonstrates for the first time that insulator complexes assembled on chromatin can attenuate the extent of replication in discrete regions of the salivary gland polyploid genome. Despite distinct cell cycle programs in dividing and endoreplicating cells (*Zielke et al., 2013*), the core biochemical composition of replisomes in both cell types is likely similar. Although the putative relationship is limited by a paucity of comparative biochemical analyses of replication factors in different cell types, related insulator-driven control mechanisms for DNA replication may be conserved in endoreplicating and mitotically dividing diploid cells. Our data thus implicates insulator/chromatin boundary elements as a critical attribute of DNA replication control. Our model suggests that delayed replication of repressed chromatin (e.g., intercalary heterochromatin) during very late S phase can

be imposed in a simple, two-component mechanism (*Figure 6C*). First, it requires that an extended genomic domain be completely devoid of functional origins of replication. The assembly and licensing of proximal pre-RC complexes can be repressed epigenetically or at the level of DNA sequence. Second, this domain is separated from flanking chromatin by a barrier element associated with an insulator complex, such as SUMM4. This structural organization is capable of preventing or delaying the entry of external forks fired from distal origins.

An important frequent feature of the partially suppressed underreplication in *mod(mdg4)* animals is its asymmetry (*Figure 7D*, *Figure 7—figure supplement 1B*), which is consistent with a unidirectional penetration of the underreplicated domain by a replication fork firing from the nearest external origin (*Figure 6C*). The SUMM4-dependent barrier may be created as a direct physical obstacle to MCM2-7 DNA-unwinding helicase or other enzymatic activities of the replisome. Alternatively, SUMM4 may inhibit the replication machinery indirectly by assembling at the insulator a DNA/chromatin structure that is incompatible with replisome translocation. This putative inhibitory structure may involve epigenetic modifications of chromatin as proposed earlier (*Gaszner and Felsenfeld, 2006*), linker histone H1 as shown previously (*Andreyeva et al., 2017*) and may also be dependent on Rif1, a negative DNA replication regulator that acts downstream of SUUR (*Munden et al., 2018*).

In conclusion, we used a newly developed MERCI approach to identify a stable stoichiometric complex termed SUMM4 that comprises SUUR, a previously known negative effector of replication, and Mod(Mdg4), an insulator protein. SUMM4 subunits cooperate to mediate transcriptional repression and chromatin boundary functions of *gypsy*-like (class 3) insulators (*Schwartz et al., 2012*) and inhibit DNA replication likely by slowing down replication fork progression through the boundary element. Thus, SUMM4 is required for coordinate regulation of gene expression, chromatin partitioning, and DNA replication timing. The insulator-dependent regulation of DNA replication offers a novel mechanism for the establishment of replication timing in addition to the currently accepted paradigm of variable timing of replication origin firing.

## Materials and methods
### Recombinant proteins
Recombinant proteins were expressed in Sf9 cells using baculovirus system (SUUR, Mod(Mdg4), EGG, and WDE), in *Escherichia coli* (ISWI, ModT antigen, and LCMS reference proteins), or obtained from EpiCypher Inc (human BRG1/SMARCA4).

### Sf9 cells
All baculovirus constructs were cloned by PCR with Q5 DNA polymerase (New England Biolabs) and ligation or Gibson assembly with NEBuilder HiFi DNA Assembly Cloning kit (New England Biolabs) into pFastBac vector (Thermo Fisher) under the control of polyhedrin promoter. All constructs were validated by Sanger sequencing. Baculoviruses were generated according to the protocol by Thermo Fisher. The baculoviruses were isolated by plaque purification, amplified three times, and their titers were measured by plaque assay. FLAG-SUUR construct was cloned from *SuUR-RA* cDNA (LD13959, DGRC). The following open-reading frame (ORF) was expressed: M**DYKDDDDK**H-SUUR-PA(1..962)-VEACGTKLVEKY*. To generate ATPase-dead mutant, SUUR-PA(K59) codon was replaced with an alanine codon by PCR and Gibson cloning. Mod(Mdg4)-67.2-V5-His$_6$ and Mod(Mdg4)-59.1-V5-His$_6$ constructs were cloned from cDNAs *mod(mdg4)-RT* and *mod(mdg4)-RI* synthesized as gBlocks by IDT, Inc. The following ORFs were expressed: Mod(Mdg4)-67.2 (1..610)-GIL**EGKPIPNPLLGLDST**GASVE-HHHHHH* and Mod(Mdg4)-59.1 (1..541)-GIL**EGKPIPNPLLGLDST**GASVEHHHHHH*. EGG-FLAG and EGG (untagged) were cloned by PCR from *egg-RA* cDNA (IP14531). The following ORF was expressed: EGG-PA(1..1262)-**DYKDDDDK*** and EGG-PA(1..1262)-*. FLAG-WDE was cloned by PCR from *wde-RA* cDNA (LD26050). The following ORF was expressed: M**DYKDDDDK**-WDE-PA(2..1420)-*. The sequences of FLAG and V5 tags are highlighted in bold typeface.

Cells, 2•10$^6$/ml in Sf-900 II SFM medium (Gibco), were infected at multiplicity of infection (MOI) of ~10 in PETG shaker flasks (Celltreat, Inc). After infection for 48–72 hr at 27°C, cells were harvested, and recombinant proteins were purified by FLAG or Ni-NTA affinity chromatography (*Fyodorov and Kadonaga, 2003*). Whereas, typically, amplified baculovirus stocks had titers above 5•10$^9$ pfu/ml, FLAG-SUUR viruses reached no more than 2–4•10$^8$ pfu/ml, presumably due to the inhibitory effect

of overexpressed protein on viral DNA replication. Accordingly, whereas typical yields of purified recombinant proteins were >100 µg from 1 L Sf9 cell culture, SUUR polypeptides were produced at no more than 2 µg from 1 L culture, which also adversely affected the protein purity (*Figures 1C and 3A*, *Figure 3—figure supplement 1A and B*).

### E. coli

The expression construct for untagged recombinant *Drosophila* ISWI was prepared from a full-length ISWI cDNA (*Ito et al., 1999*). Human TXNRD1 sequence was cloned from a cDNA provided by Addgene (#38863), and TXNRD2 was synthesized as a gBlock gene fragment by IDT, Inc. The ORFs were inserted by Gibson cloning in a pET backbone vector in frame with a C-terminal intein-CBD (chitin-binding domain) tag. Protein expression was induced by IPTG in Rosetta 2 cells, and proteins were purified in non-denaturing conditions by chitin affinity chromatography and intein self-cleavage as described (*Emelyanov et al., 2014*), followed by anion-exchange chromatography (Source 15Q) on FPLC (see below). Note that the cloned human thioredoxin reductase ORFs do not express the C-terminal selenocysteines. They were thus presumed catalytically inactive (*Arnér et al., 1999*; *Cheng and Arnér, 2017*) and designated hTXNRD1ci and hTXNRD2ci. They were used exclusively as spike-in mass standards in LCMS acquisitions of *Drosophila* proteins.

Polypeptide corresponding to the C-terminal specific region of Mod(Mdg4)-67.2 was cloned in pET24b vector in frame with a C-terminal His$_6$ tag. M-Mod(Mdg4)-67.2 (403..610)-GILEHHHHHH* was expressed in Rosetta 2 and purified by Ni-NTA affinity chromatography in non-denaturing conditions. The polypeptide (ModT) was dialyzed into PBS (137 mM NaCl, 3 mM KCl, 8 mM NaH$_2$PO$_4$, 2 mM KH$_2$PO$_4$) and used as an antigen for immunizations (see below). All recombinant proteins were examined by SDS-PAGE along with Pierce BSA mass standards (Thermo Fisher), and their concentrations were calculated from infrared scanning of Coomassie-stained gels (Odyssey Fc Imaging System, LI-COR Biosciences). Detailed cloning and purification methods are provided below.

## Molecular cloning

### pFastBac-FLAG-SUUR

The coding sequence was amplified from LD13959 by PCR using the following primers: NdeI-SUURf, TC<u>CATATG</u>TATCACTTTGTATCCGAGCAAAC and Sal1-SUURr, AA<u>GTCGAC</u>CTTGAACAGTTCCAAT CGCTTTC (NdeI and SalI restriction sites are underlined). The PCR product was digested with NdeI and SalI and ligated with the vector produced by NdeI-XhoI digestion of pFastBac-Flag-ATRX construct (*Emelyanov et al., 2010*).

### pFastBac-FLAG-SUUR(K59A)

The complete pFastBac-FLAG-SUUR construct was amplified by PCR using the following primers: SUUR-KAf, CTTGG<u>GC</u>AGGTCGCTACGGTGGCGG and SUUR-KAr, GTAGCGACCT<u>GC</u>CCAAGGCC ACTCTCATCATTCAGG (mutated residues are underlined). The linear PCR product was re-circularized by Gibson assembly.

### pFastBac-Mod(Mdg4)-67.2-V5-His$_6$

The following gBlock (MMD4-RT) was synthesized by IDT, Inc: CGAAGCGCGCGGAATTCAT**ATG**GCCG ATGACGAACAGTTTTCGCTGTGCTGGAACAACTTTAACACAAATTTGTCGGCAGGATTTCACGA GAGTCTCTGTCGGGGCGACTTGGTAGACGTCTCCTTGGCAGCAGAGGGACAAATTGTCAAGGCC CATCGTCTGGTACTCTCCGTCTGCAGCCCATTTTTTCGGAAAATGTTCACTCAGATGCCAAGCA ACACTCACGCCATAGTATTTCTGAACAATGTTAGTCACAGCGCTTTGAAAGATCTGATCCAATTTATG TATTGTGGCGAAGTGAACGTTAAGCAAGACGCATTGCCGGCATTTATCTCCACTGCAGAAAGTC TGCAAATTAAAGGATTGACCGATAACGACCCAGCTCCGCAACCCCCACAAGAGAGCTCGCCACC TCCCGCTGCGCCTCATGTGCAGCAACAGCAAATCCCAGCCCAGCGGGTGCAACGACAACAGCCG CGTGCTAGCGCCCGCTATAAAATTGAGACTGTGGATGATGGACTGGGCGACGAAAAACAAAGTA CCACTCAGATTGTTATCCAAACAACAGCTGCCCCGCAAGCAACTATTGTTCAACAACAACAGCC TCAACAAGCTGCACAACAAATACAGTCGCAACAGTTGCAGACAGGTACAACAACAACTGCAACA TTGGTAAGTACTAATAAGAGGAGTGCTCAGCGCTCGTCCCTGACGCCGGCGTCCAGTAGTGCGG GTGTTAAAAGGAGTAAGACAAGCACTAGCGCAAACGTGATGGATCCGCTGGATTCGACTACGGA GACAGGCGCAACTACAACGGCTCAACTGGTACCTCAGCAAATCACTGTACAAACATCCGTTGTC

AGCGCTGCTGAGGCGAAGCTCCATCAGCAGAGTCCCCAACAGGTTCGCCAGGAAGAGGCG
GAGTATATAGATCTGCCTATGGAGCTGCCGACCAAGTCGGAACCGGATTACTCGGAAGATCATG
GCGACGCGGCCGGTGACGCTGAGGGTACGTATGTCGAGGATGATACGTACGGTGACATGCGATA
CGACGATTCCTATTTTACAGAAAATGAGGACGCAGGCAACCAGACGGCCGCCAATACAAGCGGA
GGTGGCGTGACAGCGACCACTAGCAAAGCTGTTGTGAAACAACAGTCGCAGAACTATTCGGAGA
GTAGTTTCGTAGATACCAGTGGCGACCAAGGTAACACCGAGGCACAGGCAGCCACAAGTGCTTC
GGCGACCAAGATTCCGCCCCGGAAACGGGGTCGACCGAAAACAAAAGTTGAGGACCAGAC
CCCTAAACCTAAATTGCTtGAGAAGTTGCAGGCCGCAACACTGAACGAGGAAGCAAGTGAACCG
GCCGTATATGCGTCGACCACGAAAGGCGGTGTTAAACTGATATTTAACGGCCATTTGTTTAAATTCTC
GTTTAGGAAAGCGGATTACAGTGTCTTCCAGTGTTGTTATAGGGAGCATGGTGAAGAGTGCAAG
GTCAGGGTCGTCTGCGATCAAAAGCGTGTATTTCCTTACGAGGGTGAACACGTGCACTTCATGC
AAGCTTCCGATAAGTCCTGCCTCCCTAGTCAGTTCATGCCAGGTGAGTCCGGTGTCATTTCCAG
TTTGAGCCCATCGAAAGAGCTCTTGATGAAGAATACCACTAAGCTCGAAGAGGCGGATGATAAG
GAAGACGAAGATTTCGAAGAGTTTGAGATCCAAGAAATAGACGAGATAGAATTGGACGAACCGG
AGAAGACCCCCGCAAAGGAAGAAGAAGTTGACCCGAACGACTTTCGGGAGAAGATTAAGC
GACGGCTCCAGAAGGCCTTGCAAAACAAAAAG**AAA**GGAATTCTC<u>GAGGGTAAGCCTATCCCTAA</u>
<u>CCCTCTCCTCGGTCTCGATTCTACC</u>GGTGCTAGCGTCGAGCACCACCACCACCACCAC**TGA**GATC
CGGCTGCTAAC (sequence coding for V5 tag is underlined; translation initiation/termination codons
and codon 610 of *mod(mdg4)-RT* are shown in bold). The vector fragment was amplified by PCR from
pFastBac by using the following primers: His-Stop-Vf, CAC**TGA**GATCCGGCTGCTAAC and NdeI-Vr,
**CAT**ATGAATTCCGCGCGCTTC. The expression construct was assembled by Gibson cloning.

### pFastBac-Mod(Mdg4)-59.1-V5-His$_6$

The following gBlock (MMD4-RI) was synthesized by IDT, Inc: GGTAACACCGAGGCACAG**GTAT**
**GTGATGATCTCGATGACATGAAAGGCGCTATTAAGCATAGCCTGTTGACTTTTATTCGCGGTCA**
**GCGCGGCTGCAAACTGCTGGCTTTTAACGGTCATAATTATGTTCGTAACAGGCGTTCCAATCTC**
**AAGACGTATTGGATATGCAGCAAAAAAGGCAGCACTAAATGCAACGCTCGTGTTGTTACAAACG**
**TAGTTGAGGGTGTTCACAAGATAGTTCTGGAAAGTTGCCATCATACGTGTCTGAACACCGAGAG**
**GAAGAAAAGGCTCTCGGTGACTAATGTAGTAGGAAAAGCGCGGTCGAAGTCCGAAAAAAG**
**TGTATCCACGGGCTTTATTAAAGAAGAAGGAGACGAGGACCTCACGTTGGAATTGCGGACCCTC**
**AACCTGTCGATTGAGGATCTGAATAACCTCCAG**GGAATTCTC<u>GAGGGTAAGCC</u> (sequence corre-
sponding to V5 tag is underlined; variant-specific codons 403–541 of *mod(mdg4)-RI* are shown in
bold). The vector fragment additionally encompassing *mod(mdg4)* codons 1–402 were amplified by
PCR from pFastBac-Mod(Mdg4)-67.2-V5-His$_6$ by using the following primers: GIL-V5f, GGAATTCT
C<u>GAGGGTAAGCC</u> and MMD397-402r, **C**CTGTGCCTCGGTGTTACC. The expression construct was
assembled by Gibson cloning.

## pFastBac-EGG (untagged)

pFastBac-ATRX (untagged) construct (*Emelyanov et al., 2010*) was digested with EcoRI and XhoI. The
vector fragment (4.7 kbp) was ligated with a 4 kbp EcoRI-XhoI fragment of *egg-RA* cDNA (IP14531).

### pFastBac-EGG-FLAG

Double-stranded oligonucleotide was produced by annealing ApaI-FLAG-AflII-f, CCCAATTGCCGC
CTTCGTCTGCTC**GATTACAAGGATGATGATGACAAATAA**C and AflII-FLAG-ApaI-r, <u>TTAAG</u>***TTATTTG***
***TCATCATCATCCTTGTAATC***GAGCAGACGAAGGCGGCAATTGGG<u>GGCC</u> (sticky ends are under-
lined; sequences corresponding to FLAG tag are shown in bold; stop codon is in bold and italics) was
cloned into ApaI-AflII-digested IP14531 by ligation. The resulting construct was digested with EcoRI
and XhoI, and the 4 kbp EGG-FLAG fragment was cloned into pFastBac as described above.

### pFastBac-FLAG-WDE

pFastBac-ATRX (untagged) construct (*Emelyanov et al., 2010*) was digested with NdeI and NcoI. The
vector fragment additionally encompassing 1.1 kbp of ATRX cDNA sequence with a XhoI site (5.8 kbp
total) was ligated with a double-stranded oligonucleotide produced by annealing NdeI-FLAG-NcoI-f,
<u>T</u>**ATGGATTACAAGGATGATGATGACAAA**ATGGGAGTAAACCAGAC and NcoI-FLAG-NdeI-r, <u>CATG</u>
GTCTGGTTTACTCCCAT**TTTGTCATCATCATCCTTGTAATCCA** (sticky ends are underlined; sequences

corresponding to FLAG tag are shown in bold). A 4.6 kbp NcoI-XhoI fragment of *wde-RA* cDNA (LD26050) was cloned in the resulting construct by restriction digest and ligation.

## pET24-ISWI-intein-CBD

ISWI cDNA was amplified from pFastBac-ISWI construct (*Ito et al., 1999*) by PCR using the following primers: NdeI-ISWIf, GTTT<u>CAT</u>**ATGGCTAGCAAAACAGATAC** and XhoI-ISWIr, GGAA<u>GGTACC</u>CTTG GCAAAGCA**CCCCTTCTTCTTCTTTTTC** (NdeI and XhoI sites are underlined; sequences corresponding to the ISWI ORF are shown in bold). The 3.1 kbp PCR fragment was digested with NdeI and XhoI and cloned into pET24-intein-CBD construct in place of Protamin B (*Emelyanov et al., 2014*) by ligation.

## pET24-hTXNRD1ci-intein-CBD

Human TXNRD1 cDNA (Addgene #38863) was amplified by PCR using the following primers: NdeI-hTXNRD1f, AA<u>CAT</u>**ATGAACGGCCCTGAAGATCTTC** and SalI-hTXNRD1r, TA<u>GTCGAC</u>G **CAGCCAGCCTGGAGG** (NdeI and SalI sites are underlined; sequences corresponding to the TXNRD1 ORF are shown in bold). The 1.5 kbp PCR fragment was digested with NdeI and SalI and cloned into NdeI and XhoI sites of pET24-intein-CBD construct in place of Protamin B (*Emelyanov et al., 2014*) by ligation.

## pET24-hTXNRD2ci-intein-CBD

The following gBlock (TXNRD2) was synthesized by IDT, Inc: TTTT<u>CATATG</u>GAAGATCAGGCGGGCC AGCGCGATTATGATCTGCTGGTGGTGGGCGGCGGCAGCGGCGGCCTGGCGTGCGCGAAAG AAGCGGCGCAGCTGGGCCGCAAAGTGGCGGTGGTGGATTATGTGGAACCGAGCCCGCAGG GCACCCGCTGGGGCCTGGGCGGCACCTGCGTGAACGTGGGCTGCATTCCGAAAAAACTGA TGCATCAGGCGGCGCTGCTGGGCGGCCTGATTCAGGATGCGCCGAACTATGGCTGGGAAGTGGC GCAGCCGGTGCCGCATGATTGGCGCAAAATGGCGGAAGCGGTGCAGAACCATGTGAAAAG CCTGAACTGGGGCCATCGCGTGCAGCTGCAGGATCGCAAAGTGAAATATTTTAACATTAAAGCG AGCTTTGTGGATGAACATACCGTGTGCGGCGTGGCGAAAGGCGGCAAAGAAATTCTGCTGAGCG CGGATCATATTATTATTGCGACCGGCGGCCGCCCGCGCTATCCGACCCATATTGAAGGCGCGCT GGAATATGGCATTACCAGCGATGATATTTTTTGGCTGAAAGAAAGCCCGGGCAAAACCCTGGTG GTGGGCGCGAGCTATGTGGCGCTGGAATGCGCGGGCTTTCTGACCGGCATTGGCCTGGATACCA CCATTATGATGCGCAGCATTCCGCTGCGCGGCTTTGATCAGCAGATGAGCAGCATGGTGATTGA ACATATGGCGAGCCATGGCACCCGCTTTCTGCGCGGCTGCGCGCCGAGCCGCGTGCGCCGCCTG CCGGATGGCCAGCTGCAGGTGACCTGGGAAGATAGCACCACCGGCAAAGAAGATACCGGC ACCTTTGATACCGTGCTGTGGGCGATTGGCCGCGTGCCGGATACCCGCAGCCTGAACCTGGAAA AAGCGGGCGTGGATACCAGCCCGGATACCCAGAAAATTCTGGTGGATAGCCGCGAAGCGACCAG CGTGCCGCATATTTATGCGATTGGCGATGTGGTGGAAGGCCGCCCCGGAACTGACCCCGACCGCG ATTATGGCGGGCCGCCTGCTGGTGCAGCGCCTGTTTGGCGGCAGCAGCGATCTGATGGATTATG ATAACGTGCCGACCACCGTGTTTACCCCGCTGGAATATGGCTGCGTGGGCCTGAGCGAAGAAGA AGCGGTGGCGCGCCATGGCCAGGAACATGTGGAAGTGTATCATGCGCATTATAAACCGCTGGAA TTTACCGTGGCGGGCCGCGATGCGAGCCAGTGCTATGTGAAAATGGTGTGCCTGCGCGAACCGC CGCAGCTGGTGCTGGGCCTGCATTTTCTGGGCCCGAACGCGGGCGAAGTGACCCAGGGCTTTGC GCTGGGCATTAAATGCGGCGCGAGCTATGCGCAGGTGATGCGCACCGTGGGCATTCATCCGACC TGCAGCGAAGAAGTGGTGAAACTGCGCATTAGCAAACGCAGCGGCCTGGATCCGACCGTGACCG GC**TGC**<u>CTCGAG</u>TTTTTTTTTTT (NdeI and XhoI sites are underlined; translation initiation codon and codon 492 of hTXNRD2 are shown in bold). The DNA fragment was digested with NdeI and XhoI and cloned by ligation in pET24-intein-CBD as described above.

## pET24-ModT-His$_6$

Mod(Mdg4)-67.2-specific fragment of *mod(mdg4)-RT* cDNA was amplified from pFastBac-Mod(Mdg4)-67.2-V5-His$_6$ by PCR using the following primers: NdeI-ModTf, CCGAG<u>CATATG</u>GCAGCCACAAGT GCTTC and XhoI-ModTr, GGGTAGGCTTACC<u>CTCGAG</u>AATTCCTTTC (NdeI and XhoI sites are underlined). The 0.6 kbp PCR fragment was digested with NdeI and XhoI and cloned in pET24b (Millipore/Sigma) by ligation.

## FPLC purification of recombinant ISWI, hTXNRD1ci, and hTXNRD2ci

Protein samples eluted from the chitin resin (1–5 ml total sample volume) were diluted threefold with chromatographic Buffer A (*Figure 1—source data 1*) and injected on a 0.5 ml Source 15Q equilibrated to 5% Buffer B (*Figure 1—source data 1*) + 95% Buffer A. The column was washed with 20 *cv* (column volumes) of 5% Buffer B, and proteins were eluted with a 20 *cv* linear gradient of 5–100% Buffer B. 200 µl fractions were collected and analyzed by SDS-PAGE. Three to five peak fractions were pooled, aliquoted, flash-frozen in liquid nitrogen, and stored at –80°C.

## Crude cell extracts

### Nuclear extract from *Drosophila* embryos

Approximately 1 kg or approximately 200 g wild-type (Oregon R) *Drosophila* embryos were collected 0–12 hr after egg deposition (AED) from population cages. The embryos were dechorionated, and nuclear extracts were prepared as described (*Kamakaka et al., 1991*). Protein concentration was measured by Pierce BCA assay (Thermo Fisher). The extracts were fractionated by FPLC (*Figure 1A*, *Figure 4—figure supplement 2A*) on AKTA PURE system (Cytiva Life Sciences). Aliquots of chromatographic fractions were examined by quantitative shotgun proteomics or Western blot analyses as described below. Peak SUUR or Mod(Mdg4) fractions were diluted to an appropriate ionic strength (if applicable) and used as a starting material for the next chromatographic step. Details on FPLC column sizes and run parameters are shown in *Figure 1—source data 1* and *Figure 4—figure supplement 2—source data 1*.

### *E. coli* lysate

A 40 ml Rosetta 2 overnight culture was harvested by centrifugation, resuspended in 20 ml HEG (25 mM HEPES, pH 7.6, 0.1 mM EDTA, 10% glycerol) supplemented with 0.1 M KCl, 1 mM DTT, and 2 mM $CaCl_2$. Cells were disrupted by sonication and centrifuged to remove insoluble material. Nucleic acids were digested with 15 units micrococcal nuclease (Sigma-Aldrich) for 20 min at 37°C, and the proteins were precipitated with 2 M ammonium sulfate. The pellet was resuspended in 10 ml HEG + 0.1 M KCl +1 mM DTT with protease inhibitors (0.5 mM benzamidine, 0.2 mM PMSF) and dialyzed against the same buffer. After centrifugation, the concentration of soluble protein was measured by BCA assay, the *E. coli* lysate was diluted to 1 mg/ml using 100 mM ammonium bicarbonate (ABC) and stored at –80°C.

## Mass-spectroscopy samples

### Column fractions

For each chromatographic step, 14–20 fractions were selected based on the protein fractionation profile according to the UV ($A_{280}$) absorbances measurements. 50–100 µl aliquots of chromatographic fractions, starting material (SM) and column flow-through (FT, if applicable) were saved, and protein concentrations were estimated based on their UV absorbances (1000 mU $A_{280}$ was considered to be equivalent to 5 mg/ml total protein). Equal volumes of each fraction, SM, and FT were used for MS acquisitions, so that no more than 40 µg total protein was processed in each reaction. As a reference, the reactions were supplemented with 1.5 µg each of purified recombinant human thioredoxin reductases 1 and 2 (hTXNRD1ci and hTXNRD2ci, catalytically inactive) expressed in *E. coli*. Dithiotreitol (DTT) was added to the protein samples to 10 mM and NP-40 – to 0.02%. Reaction volumes were brought to 85 µl with 50 mM ABC. All reagents, including water, were HPLC/MS grade. The proteins were reduced for 1 hr at 37°C and then alkylated with 30 mM iodoacetamide (IAA, 15 µl 200 mM IAA in water) for 45 min at room temperature in the dark. Alkylated proteins were desalted into 50 mM ABC using ZebaSpin columns (40 kDa MWCO) and digested with 1 µg trypsin for 2 hr at 37°C. 1 µg more trypsin was added, and the digestion progressed at 37°C overnight. Tryptic peptides were lyophilized for 2 hr on SpeedVac with heat and resuspended in 100 µl Sample Buffer: 1% acetonitrile (ACN) and 0.1% formic acid (FA) in water. Equal volumes (23 µl) of samples were used for IDA and SWATH acquisitions (in triplicate) as described below.

## Recombinant SUUR

To generate the recombinant SUUR reference spectral library (ILR), ~0.5 µg purified recombinant FLAG-SUUR (both 130 and 65 kDa bands, *Figure 1C*) was mixed with 1.5 µg each of hTXNRD1ci and

hTXNRD2ci and processed for an IDA as described above, except for 0.5 µg trypsin was used in each cleavage step, and the peptide sample was resuspended in 30 µl Sample Buffer. For SWATH titration of SUUR (*Figure 1—figure supplement 1B*), 1 µg recombinant FLAG-SUUR was mixed with 25 µg *E. coli* lysate protein and 1.5 µg each of hTXNRD1ci and hTXNRD2ci. Tenfold serial dilutions down to 10 fg SUUR were also prepared using the mixture of *E. coli* lysate with reference proteins. The samples were processed for SWATH acquisitions in triplicate as described above, 30 µl of sample per injection.

## In-gel digestion of recombinant proteins for LCMS identification

Recombinant SUUR or SUMM4 purified by FLAG immunoaffinity chromatography was resolved on SDS-PAGE, stained with Coomassie Blue (*Figures 1C and 3A*, *Figure 3—figure supplement 1A*), and up to eight most prominent protein bands were excised. The gel slices were transferred to 1.5 ml Eppendorf tubes, gently crushed with a RotoDounce pestle, and destained with 25 mM ABC in 50% methanol and then with 25 mM ABC in 50% ACN (30 min each at room temperature). The proteins were reduced in 50 µl 10 mM DTT for 1 hr at 55°C and alkylated with 30 mM IAA for 45 min at room temperature in the dark. The gel fragments were washed with 25 mM ABC in 50% ACN, dehydrated with 100% ACN, dried in a SpeedVac, rehydrated by addition of 50 µl 50 mM ABC, and digested with 0.25 µg trypsin overnight at 37°C. The peptides were extracted once with 50 µl 10% FA and once with 100 µl 3% FA in 60% ACN, both extracts were combined, dried in a SpeedVac and resuspended in 50 µl Sample Buffer. Then, 40 µl of each sample was injected for IDA as described below.

## Mass-spectroscopy acquisition methods

LC-MS/MS analyses were performed on a TripleTOF 5600+ mass spectrometer (AB SCIEX) coupled with M5 MicroLC system (AB SCIEX/Eksigent) and PAL3 autosampler.

### Instrument settings

LC separation was performed in a trap-elute configuration, which consists of a trapping column (LUNA C18(2), 100 Å, 5 µm, 20 × 0.3 mm cartridge, Phenomenex) and an analytical column (Kinetex 2.6 µm XB-C18, 100 Å, 50 × 0.3 mm microflow column, Phenomenex). The mobile phase consisted of water with 0.1% FA (phase A) and 100% ACN containing 0.1% FA (phase B). Then, 200 ng to 10 µg total protein was injected for each acquisition. Peptides in Sample Buffer were injected into a 50 µl sample loop, trapped and cleaned on the trapping column with 3% mobile phase B at a flow rate of 25 µl/ min for 4 min before being separated on the analytical column with a gradient elution at a flow rate of 5 µl/min. The gradient was set as follows: 0–48 min: 3% to 35% phase B, 48–54 min: 35% to 80% phase B, 54–59 min: 80% phase B, 59–60 min: 80% to 3% phase B, and 60–65 min at 3% phase B. An equal volume of each sample (23 µl) was injected four times, once for information-dependent acquisition (IDA), immediately followed by DIA/SWATH in triplicate. Acquisitions of distinct samples were separated by a blank injection to prevent sample carryover. The mass spectrometer was operated in positive ion mode with EIS voltage at 5200 V, Source Gas 1 at 30 psi, Source Gas 2 at 20 psi, Curtain Gas at 25 psi, and source temperature at 200°C.

### IDA and data analyses

IDA was performed to generate reference spectral libraries for SWATH data quantification. The IDA method was set up with a 250 ms TOF-MS scan from 400 to 1250 Da, followed by MS/MS scans in a high-sensitivity mode from 100 to 1500 Da of the top 30 precursor ions above 100 cps threshold (100 ms accumulation time, 100 ppm mass tolerance, rolling collision energy, and dynamic accumulation) for charge states ($z$) from +2 to +5. IDA files were searched using ProteinPilot (version 5.0.2, ABSciex) with a default setting for tryptic digest and IAA alkylation against a protein sequence database. The *Drosophila* proteome FASTA file (21,970 protein entries, UniProt UP000000803, 3/21/2020) augmented with sequences for common contaminants as well as hTXNRD1 and hTXNRD2 was used as a reference for the search. Up to two missed cleavage sites were allowed. Mass tolerance for precursor and fragment ions was set to 100 ppm. A false discovery rate (FDR) of 5% was used as the cutoff for peptide identification.

## SWATH acquisitions and data analyses

For SWATH (SWATH-MS, Sequential Window Acquisition of All Theoretical Mass Spectra) acquisitions (*Zhu et al., 2014*), one 50 ms TOF-MS scan from 400 to 1250 Da was performed, followed by MS/MS scans in a high-sensitivity mode from 100 to 1500 Da (15 ms accumulation time, 100 ppm mass tolerance, +2 to +5 $z$, rolling collision energy) with a variable-width SWATH window (*Zhang et al., 2015*). DIA data were quantified using PeakView (version 2.2.0.11391, ABSciex) with SWATH Acquisition MicroApp (version 2.0.1.2133, ABSciex) against selected spectral libraries generated in Protein-Pilot. Retention times for individual SWATH acquisitions were calibrated using 20 or more peptides for hTXNRD1ci and hTXNRD2ci. The following software settings were utilized: up to 25 peptides per protein, 6 transitions per peptide, 95% peptide confidence threshold, 5% FDR for peptides, XIC extraction window 20 minutes, and XIC width 100 ppm. Protein peak areas were exported as Excel files (*Supplementary file 2*) and processed as described below.

## MERCI

MERCI is a novel approach for rapid identification of native protein complexes. It combines enrichment for a target subunit of a putative complex by consecutive FPLC steps and quantitative shotgun proteomics of chromatographic fractions. Crude nuclear extract from *Drosophila* embryos was fractionated as in *Figure 1A* and *Figure 1—source data 1*. At every step, 40 μg or less total protein from each of 10–20 fractions (equal volumes) was supplemented with a fixed amount (1.5 μg each) of exogenous reference proteins (human thioredoxin reductases), reduced, alkylated, and digested with trypsin (see above). MS1 and MS2 spectra of tryptic peptides were acquired by IDA, and relative SUUR abundance in fractions was measured by DIA/SWATH in triplicate. SWATH data were quantified using cognate IDA-derived ion libraries. Protein areas for all quantified proteins were normalized to the sum of those for reference proteins. The relative numbers were averaged across triplicates, with standard deviations calculated. The average numbers for all quantified proteins were further normalized by converting them to Z-scores (see *Supplementary file 2* for an example of calculations). Peak SUUR fractions (1–5) were then subjected to the next FPLC/MERCI step. After five column steps, the IL from the ultimate FPLC step (IL5) was used to requantify SWATH data from all steps. Z-scores for all purification steps were stitched together, and the large array encompassing all data points for every protein was analyzed by Pearson correlation with SUUR (*Supplementary file 2*). The most closely correlated purification profiles served as an indication for protein co-purification, potentially, as subunits of a stable complex.

## Biochemical assays with recombinant proteins

### Oligonucleosome substrates

Oligonucleosomes were reconstituted in vitro as described (*Lu et al., 2013*) from supercoiled plasmid DNA (3.2 kb, pGIE-0), native core histones and H1 prepared from *Drosophila* embryos (*Fyodorov and Levenstein, 2002*) by gradient salt dialysis in the presence of 0.2 mg/ml nuclease-free bovine serum albumin (BSA, New England Biolabs). Quality of reconstitution was assessed by SDS-PAGE (*Figure 3—figure supplement 1C*), MNase (*Figure 3—figure supplement 1D*), and chromatosome stop assays (*Figure 3—figure supplement 1E*).

### ATPase assay

40 nM recombinant proteins were incubated in 25 μl reaction buffer containing 20 mM HEPES, pH 7.6, 0.15 M NaCl, 4 mM $MgCl_2$, 1 mM ATP, 0.1 mM EDTA, 0.02% (v/v) NP-40, and 0.1 mg/ml nuclease-free BSA for 60 min at 27°C. Some reactions additionally contained 10 nM pGIE-0 plasmid DNA or equivalent amounts of oligonucleosomes ± H1. ATPase assays were performed using ADP-Glo Max kit (Promega). All reactions were performed in triplicate, the results were normalized to the ADP-ATP titration curve according to the kit manual and converted to enzymatic rates (molecules of ATP hydrolyzed per molecule of enzyme per minute). Averages and standard deviations were calculated. Statistical differences were calculated by Mann–Whitney test.

## EpiDyne-PicoGreen nucleosome remodeling assay

EpiDyne-PicoGreen is a restriction enzyme accessibility assay modified for increased throughput and sensitivity (*Figure 3—figure supplement 2A*). Briefly, a recombinant ATPase over a concentration range (*Figure 3—figure supplement 2B–E*) was mixed with 10 nM EpiDyne biotinylated nucleosome remodeling substrate (EpiCypher), terminally positioned 6-N-66 (217 bp fragment) or centrally positioned 50-N-66 (263 bp) and 1 mM ATP in 20 µl remodeling buffer, 20 mM Tris-HCl, pH 7.5, 50 mM KCl, 3 mM $MgCl_2$, 0.01% (v/v) Tween-20, 0.01% (w/v) BSA. The remodeling reactions were incubated at 23°C in 384-well format. At indicated time points, the reactions were quenched, and nucleosome substrates were immobilized on an equal volume of streptavidin-coated magnetic beads (NEB), pre-washed and resuspended in 2× quench buffer, 20 mM Tris-HCl, pH 7.5, 600 mM KCl, 0.01% (v/v) Tween-20, and 0.01% (w/v) BSA. Beads were successively washed by collection on a magnet (three times with wash buffer, 20 mM Tris-HCl, pH 7.5, 300 mM KCl, 0.01% [v/v] Tween-20) and buffer replacement (once with RE buffer, 20 mM Tris-HCl, pH 7.5, 50 mM KCl, 3 mM $MgCl_2$, 0.01% [v/v] Tween-20). Beads were resuspended in 20 µl restriction enzyme mix, 50 units/ml Dpn II (NEB) in RE buffer, and incubated at 23°C for 30 min, collected on a magnet, and supernatants from all wells were transferred to a new plate. They were mixed with an equal volume of Quant-iT PicoGreen dsDNA reagent (Thermo Fisher, Component A) and 1 unit/ml thermolabile proteinase K (NEB) in TE and incubated at 23°C for 1 hr. Fluorescence intensity was detected on an EnVision microplate reader with excitation at 480 nm and emission at 531 nm, and data expressed as relative fluorescence units (RFUs) through the EnVision Workstation (version 1.13.3009.1409).

## *Drosophila* population culture, mutant stocks, and genetics

Wild-type (Oregon R) flies were maintained in population cages on agar-grape juice and yeast paste plates at 26°C, 60% humidity with 12 hr dark–light cycle. Mutant flies were reared, and crosses were performed at 26°C on standard cornmeal/molasses medium with dry yeast added to the surface. $SuUR^{ES}$ was a gift of Igor Zhimulev, and $mod(mdg4)^{m9}$ was a gift of Yuri Schwartz. All other alleles were obtained from the Bloomington Stock Center, Indiana. Combinations of alleles were produced either by crosses with appropriate balancers and segregation of markers or by female germline meiotic recombination. Intra-chromosomal recombination events were confirmed by PCR of genomic DNA. To genotype $SuUR^{ES}$, $mod(mdg4)^{u1}$ recombined chromosomes, the following PCR primers were used: SUUR-Fwd: CCTCAAAGAACAGCCAGAGC; SUUR-Rev: TTTGCTACTTCTGGGCGTTT; diver-Rev: TCAGTTTGAACTCGCACCAG; Mod-Fwd: CAGGGCCACACGCACTTAC; Mod-Rev: GTGAAGCC CTTAGGCAGCTC; and Stalker-Rev: GCTTGCAGCACAGTTAGCAC. SUUR-Fwd/SUUR-Rev combination of primers produced a 770 bp PCR product for wild-type $SuUR$. SUUR-Fwd/diver-Rev combination produced an ~850 bp PCR product for $SuUR^{ES}$. Mod-Fwd/Mod-Rev combination produced a 1532 bp PCR product for wild-type $mod(mdg4)$. Mod-Fwd/Stalker-Rev combination produced an ~1700 bp PCR product for $mod(mdg4)^{u1}$.

Fly wings were dissected from ~5-day-old adult males and transferred to a drop of PBS + 0.1% Triton X-100 (PBST). The wings were soaked in 80% glycerol in PBST and photographed using Zeiss AxioVert 200M microscope with EC Plan-Neofluar 2.5×/0.075 lens in bright field and CCD monochrome camera AxioCam MRm. For wing area measurements, images were processed using Fiji/ImageJ2 software package. Statistical differences were calculated by two-tailed *t*-test, assuming unequal variances. Adult fly eye images were taken on live, $CO_2$-anesthetized 2-day-old females on Zeiss stereomicroscope Discovery.V12 using CCD color camera AxioCam MRc.

## Antibodies, immunoblots, and immunoprecipitation (IP)

Polyclonal antibody (anti-ModT) was raised in guinea pigs by Pocono Rabbit Farm & Lab. Rabbit polyclonal antibody to the C-terminus of *Drosophila* XNP/ATRX (anti-XNP) was described previously (*Emelyanov et al., 2010*). Rabbit and guinea pig polyclonal antibodies to *Drosophila* SUUR were a gift of Alexey Pindyurin (*Nordman et al., 2014*) and Igor Zhimulev (*Pindyurin et al., 2008*). Rabbit polyclonal Mod(Mdg4)-FL antibody to full-length Mod(Mdg4)-67.2 that recognizes all splice forms of Mod(Mdg4) was a gift of Jordan Rowley and Victor Corces. Mouse monoclonal anti-FLAG (M2, Sigma Aldrich), anti-PCNA (PC10, Cell Signaling), anti-β-tubulin and anti-HP1a (E7 and C1A9, Developmental Studies Hybridoma Bank) were obtained commercially.

Western blotting was performed using standard techniques. For FPLC column fraction analyses, 5–10 μl of starting material and flow-through (if applicable) and 5–15 μl of column fractions were loaded per lane. For expression analyses in salivary glands, 10 salivary glands from L3 larvae of indicated genotype were frozen and thawed, boiled extensively in 40 μl 2× SDS-PAGE loading buffer, centrifuged, and the material equivalent to four salivary glands was loaded per lane. The following dilutions were used: 1:200,000 anti-ModT, 1:1,000 anti-Mod(Mdg4)-FL, 1:1000 guinea pig and rabbit anti-SUUR, 1:1000 anti-HP1a, 1:1000 anti-β-tubulin, and 1:2000 anti-FLAG. Infrared-labeled secondary antibodies: donkey anti-guinea pig IRDye 800CW, goat anti-mouse IRDye 800CW, goat anti-rabbit IRDye 800CW, goat anti-rabbit IRDye 680CW, and goat anti-mouse IRDye 680RD were obtained from Li-COR Biosciences and used at 1:10,000. The blots were scanned on Odyssey Fc Imaging System (LI-COR Biosciences).

Immunoprecipitation experiments were performed as described (*Emelyanov et al., 2012*). 400 μl *Drosophila* embryonic nuclear extracts (~10 mg total protein) were incubated with 10 μl guinea pig anti-ModT, 30 μl rabbit anti-SUUR, or 20 μl rabbit anti-XNP antibodies for 3 hr at 4°C. Immunocomplexes were collected by addition of 25 μl protein A-agarose plus (Thermo Fisher) for 2 hr at 4°C. After washing four times with 1 ml of buffer HEG (25 mM HEPES, pH 7.6, 0.1 mM EDTA, 10% glycerol) + 0.15 M NaCl, the immunoprecipitated proteins were eluted with 80 μl 2× SDS-PAGE loading buffer and analyzed by SDS-PAGE and Western blot using guinea pig or rabbit anti-SUUR and anti-Mod(Mdg4) and mouse anti-HP1a antibodies. For Mod(Mdg4) and HP1a, 8 μl of immunoprecipitated material (equivalent to 1 mg nuclear extract proteins) and 5% input (2 μl nuclear extract, 50 μg total protein) were analyzed. For SUUR, 20 μl of immunoprecipitated material (equivalent to 2.5 mg nuclear extract proteins) and 10% input (10 μl nuclear extract, 250 μg total protein) were analyzed.

## Polytene chromosomes and indirect IF analyses

For all cytological experiments, larvae were reared and collected at 18°C. Polytene chromosomes and whole-mount salivary glands were prepared and analyzed as described previously (*Andreyeva et al., 2017*). Briefly, salivary glands from wandering third-instar larvae were dissected in PBS. Glands were transferred into a formaldehyde-based fixative (one ~15 μl drop of 3% lactic acid, 45% acetic acid, 3.7% formaldehyde on a coverslip) for 2 min, squashed, and frozen in liquid N₂. The coverslips were removed, and slides were placed in 70% ethanol for 20 min and stored at −20°C. The slides were washed three times for 5 min in PBST. Primary antibodies were incubated overnight at 4°C in PBST + 0.1% BSA and washed three times for 5 min each with PBST. Secondary antibodies were incubated for 2 hr at room temperature in PBST + 0.1% BSA and washed three times for 5 min each with PBST.

DNA was stained with 0.1 μg/ml DAPI in PBST for 3 min, and squashes were mounted in Prolong Glass anti-fade mountant (Molecular Probes). Primary and secondary antibodies were used at the following dilutions: guinea pig anti-ModT, 1:50,000; rabbit anti-SUUR, 1:100; mouse anti-PCNA, 1:1000; mouse anti-FLAG, 1:100; Alexa Fluor 488 highly cross-absorbed (HCA) goat anti-mouse, Alexa Fluor 568 HCA goat anti-guinea pig, and Alexa Fluor 647 plus HCA goat anti-rabbit (all Thermo Fisher), all 1:800. IF images were obtained with Zeiss AxioVERT 200M microscope and AxioCam MRm mono microscopy camera using a ×40/1.3 Plan-Neofluar or ×63x/1.40 Plan-Apochromat lenses with oil immersion. Images were acquired using AxioVision software.

For whole-mount IF staining, L3 larvae were reared at 26°C, and salivary glands were dissected in PBS and fixed in 3.7% formaldehyde (Sigma-Aldrich) for 20 min at room temperature. The glands were washed in PBS + 0.3% Triton X-100 and permeabilized for 30 min at 37°C in PBS + 1% Triton X-100. Blocking was performed for 30 min at room temperature in PBS + 0.3% Triton X-100 supplemented with 10% fetal calf serum and 1% BSA. The glands were incubated with primary antibodies diluted in blocking solution for 48 hr at 4°C, washed three times with PBS + 0.3% Triton X-100 for 30 min, and incubated with secondary antibodies in blocking solution overnight at 4°C. The stained glands were washed three times with PBS + 0.3% Triton X-100 for 30 min, stained with DAPI (0.1 μg/ml) for 30 min, and mounted in Prolong Gold anti-fade (Invitrogen). IF images were obtained on a Leica SP8 confocal microscope using a ×20/0.75 PLAPO lens and processed using Fiji/ImageJ software.

To quantify the putative colocalization of SUUR and Mod(Mdg4)-67.2 in polytene chromosomes (*Figure 4A*), the image resolution was reduced to 1388 by 1040. Pixel intensities (1,443,520) for SUUR and ModT channels were extracted from Bitmap files (ImageJ), normalized to Z-scores, and plotted as an X-Y scatter plot (*Figure 4—figure supplement 1C*). For colocalization analyses, the plot regions

($Z_{ModT} > 1$ and $Z_{SUUR} < 3$, green) and ($Z_{ModT} < 1$ and $Z_{SUUR} > 3$, red) were excluded from consideration (*Figure 4—figure supplement 1D*).

## Next-generation sequencing analyses (NGS)

Salivary glands from female wandering third-instar larvae were isolated and flash-frozen in liquid $N_2$ until all samples were collected. Genomic DNA for sequencing was prepared from 25 L3 salivary gland pairs or 10 mg embryos (0–6 hr AED) using DNeasy Blood and Tissue kit (QIAGEN). Each sample was prepared in triplicate. The tissues were soaked in 180 µl buffer ATL + 20 µl proteinase K (15 mg/ml) and lysed for 2–3 hr at 55°C. The reactions were cooled to room temperature, supplemented with 4 µl RNase A, ~40 mg/ml (Sigma-Aldrich), and RNA was digested for 10 min. The genomic DNA was fragmented with 0.002 units DNase I (Thermo Fisher) in 100 µl reactions containing 10 mM Tris-HCl, pH 7.5, 10 mM $MnCl_2$, 0.1 mM $CaCl_2$, 0.1 mg/ml RNase A, and 0.2 mg/ml nuclease-free BSA (1× reaction buffer) for 15 min at 37°C. (DNAse I dilutions were prepared using 1× reaction buffer.) Reactions were stopped by adding 5 µl 0.5 M EDTA, and DNase I was inactivated for 20 min at 65°C. The fragmented DNA was purified on QiaQuick columns using PCR purification kit (QIAGEN) and eluted in 40 µl 10 mM Tris-HCl, pH 8.0. The size distribution of DNA fragments (200–600 bp, average ~400 bp) was confirmed and DNA concentration was measured on 2100 BioAnalyzer (Agilent). Libraries were prepared from 20 ng of fragmented genomic DNA with the ThruPLEX DNA-seq kit using SMARTer DNA Unique Dual Indexes (TakaraBio) and sequenced 150 bp paired-end reads on an NovaSeq 6000 (Novagene).

The sequencing quality of each sample was assessed using FASTQC version 0.11.7 (*Andrews, 2010*). Raw paired-end reads were trimmed of adapters using BBDuk from the BBTools software version 38.71 using the parameters: `ktrim=r ref=adapters rcomp=t tpe=t tbo=t hdist=1 mink=11` (*Bushnell, 2014*). Reads were aligned to the BDGP Release 6 of the *Drosophila melanogaster* genome (dm6) (*dos Santos et al., 2015*) using Bowtie2 version 2.3.4.1 (*Langmead and Salzberg, 2012*) and parameters `-q --local --very-sensitive-local --no-unal --no-mixed --no-discordant --phred33 -I 10 -X 700`. Duplicate reads were marked using Picard 2.2.4 (*BroadInstitute, 2020*) and SAM files were converted to BAM format, filtered for quality (`-bq 5`), and removed of duplicates (`-bF 0x400`) using Samtools version 1.9 (*Danecek et al., 2021*). To examine replicate concordance, a principal component analysis (PCA) was performed using the deepTools package. Replicates clustered indicating high genome-wide similarity within genotypes (*not shown*). For visualization, replicates were merged (`samtools merge`) and coverage was calculated across 50 bp bins and normalized to counts per million (CPM) using deeptools version 3.2.0: `bamCoverage -bs 50 –normalizeUsing CPM` (*Ramírez et al., 2016*). Each genotype was scaled to the diploid Oregon R embryo signal in 5 kb bins: `bigWigCompare --operation first –bs 5000`. DamID-chip data for SUUR and Su(Hw) were retrieved from GSE22069 (*Filion et al., 2010*). ChIP-chip data for Su(Hw) insulator elements were also used (*Nègre et al., 2010*). Underreplicated domains were called using a custom R script to identify regions at least 100 kb in length that fell below the average chromosomal read count as described (*Andreyeva et al., 2017*). Visualization of all data was performed on the UCSC Genome browser using the dm6 release of the *Drosophila* genome (*Kent et al., 2002*). Each data set was auto-scaled to its own min and maximum, and the data were windowed by mean with 16-pixel smoothing applied.

## Quantitative real-time PCR

Genomic DNA samples prior to DNase I fragmentation (see above) were diluted to ~0.25 ng/µl. Real-time PCR was performed using 0.5 ng genomic DNA on a ViiA7 thermocycler (Applied Biosystems) with a three-step protocol (95°C 15 s, 60°C 30 s, 68°C 60 s) and iTaq Universal SYBR Green Supermix (Bio-Rad). Primer sequences are provided in *Figure 7—source data 1*. Each reaction was performed in three technical replicates for each of the three biological samples (N = 9). For each amplicon, the average Ct value (`<Ct>`) was calculated and normalized to the average Ct value for a random intergenic genomic sequence as a loading control. Further, for each template, the ΔCt was normalized to the average Ct value for embryonic DNA (diploid control). Standard deviation ($\sigma_{Ct}$) for each reaction in triplicate was also calculated. The following ΔΔCt formula was used: `<ΔΔCt> = (<Ct`target`> - <Ct`intergenic86D`>)` SG `- (<Ct`target`> - <Ct`intergenic86D`>)` embryo. Standard deviations for `<ΔΔCt>` were calculated as

$\sigma_{\Delta\Delta Ct}$ = `square root of` $(\sigma^2_{target} + \sigma^2_{intergenic86D})/2$. $\Delta\Delta Ct$'s were converted to DNA copy numbers as $2^{-<\Delta\Delta Ct>}$. The confidence interval was calculated in the range between $2^{-<\Delta\Delta Ct>-\sigma}$ and $2^{-<\Delta\Delta Ct>+\sigma}$.

To examine the putative zygotic function(s) of *SuUR*, heterozygous *SuUR^ES* parents were produced by balancing with *TM6B, Tb,* and crossed *inter se*. L3 salivary glands were dissected from homozygous *SuUR* mutant progeny, and DNA copy numbers were measured by qPCR as described above.

## Acknowledgements

This article is dedicated to the memory of Jonathan R Warner, who participated in initial discussions that have led to the development of MERCI technology. We thank B Bartholdy, K Beirit, B Birshtein, V Elagin, M Gamble, T Kolesnikova, A Lusser, A Pindyurin, C Schildkraut, Y Schwartz, S Sidoli, and I Zhimulev for helpful discussions and critical reading of the manuscript. We thank Y Schwartz, I Zhimulev, and Bloomington Stock Center for fly stocks and V Corces, A Pindyurin, J Rowley, and I Zhimulev for antibodies. We are grateful to A Aravin and B Godneeva for cloning EGG and WDE baculovirus constructs. We thank A Kumar and N Baker for help with confocal microscopy, and M Rogers and J Secombe for the use of Zeiss Discovery.V12. Confocal images were obtained at the Analytical Imaging Facility (Einstein). We thank P Schultes for help with maintaining the LCMS instrument.

## Additional information

### Competing interests

Lu Sun: Lu Sun is employed by Epicypher, Inc, a commercial developer and supplier of the EpiDyne nucleosomes and associated remodeling assay platforms used in this study. Michael-C Keogh: Michael C Keogh is employed by Epicypher, Inc, a commercial developer and supplier of the EpiDyne nucleosomes and associated remodeling assay platforms used in this study. The other authors declare that no competing interests exist.

### Funding

| Funder | Grant reference number | Author |
| --- | --- | --- |
| National Institutes of Health | R01 GM074233 | Dmitry V Fyodorov |
| National Institutes of Health | R01 GM129244 | Arthur I Skoultchi |
| National Institutes of Health | R01 GM124201 | Robert J Duronio |
| National Institutes of Health | R44 GM123869 | Michael-C Keogh |
| National Institutes of Health | T32 CA217824 | Markus Nevil |
| National Institutes of Health | K12 GM000678 | Markus Nevil |

The funders had no role in study design, data collection and interpretation, or the decision to submit the work for publication.

### Author contributions

Evgeniya N Andreyeva, Lu Sun, Formal analysis, Validation, Investigation, Visualization, Methodology, Writing – review and editing; Alexander V Emelyanov, Conceptualization, Formal analysis, Validation, Investigation, Visualization, Methodology, Writing – review and editing; Markus Nevil, Data curation, Software, Formal analysis, Funding acquisition, Validation, Visualization, Methodology, Writing – review and editing; Elena Vershilova, Christina A Hill, Investigation, Methodology; Michael-C Keogh, Resources, Formal analysis, Supervision, Funding acquisition, Validation, Visualization, Methodology, Project administration, Writing – review and editing; Robert J Duronio, Arthur I Skoultchi, Resources, Formal analysis, Supervision, Funding acquisition, Visualization, Methodology, Project administration,

Writing – review and editing; Dmitry V Fyodorov, Conceptualization, Resources, Data curation, Software, Formal analysis, Supervision, Funding acquisition, Validation, Investigation, Visualization, Methodology, Writing - original draft, Project administration, Writing – review and editing

**Author ORCIDs**
Michael-C Keogh ORCID http://orcid.org/0000-0002-2219-8623
Dmitry V Fyodorov ORCID http://orcid.org/0000-0002-3080-1787

**Decision letter and Author response**
Decision letter https://doi.org/10.7554/eLife.81828.sa1
Author response https://doi.org/10.7554/eLife.81828.sa2

## Additional files

### Supplementary files

• Supplementary file 1. Protein identities and peptide spectral data (ion libraries) obtained by information-dependent acquisitions (IDA) for FPLC fractions (IL1-5, *Figure 1A*) and recombinant SUUR (ILR, *Figure 1C*).

• Supplementary file 2. Raw data of SWATH acquisitions for FPLC fractions (*Figure 1A*) quantified using ion library IL5 (*Supplementary file 1*) and an example of protein purification profile analyses (hydroxylapatite step, *Figure 1A and H*).

• MDAR checklist

### Data availability

NGS data has been submitted to Gene Expression Omnibus (GEO, accession number GSE189421).

The following previously published dataset was used:

| Author(s) | Year | Dataset title | Dataset URL | Database and Identifier |
|---|---|---|---|---|
| Andreyeva EN, Emelyanov AV, Nevil M, Duronio RJ, Fyodorov DV | 2021 | SUMM4 complex couples insulator function and DNA replication timing control | http://www.ncbi.nlm.nih.gov/geo/query/acc.cgi?acc=GSE189421 | NCBI Gene Expression Omnibus, GSE189421 |

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
