## [Editor Report]

This important paper will be of interest to those studying DNA replication in the context of chromatin and development and to those interested in higher-order chromatin organization. It uncovers a new interaction partner for SuUR and reports how this complex (SUMM4; Suppressor of Underreplication – Modifier of Mdg4) functions to control under-replication. The results are convincing and support the conclusions.

---

## [Decision Letter]

**Decision letter after peer review:**

Thank you for submitting your article "*Drosophila* SUMM4 complex couples insulator function and DNA replication timing control" for consideration by *eLife*. Your article has been reviewed by 2 peer reviewers, and the evaluation has been overseen by a Reviewing Editor and Jessica Tyler as the Senior Editor. The reviewers have opted to remain anonymous.

Essential revisions:

1) The authors should address the comments of reviewer 1 about the role in controlling the replication timing and under-replication that occurs in polytene chromosomes. Some approaches are suggested.

2) The generality of the proposed mechanism for control of replication timing needs to be discussed, particularly whether this potential mechanism of copy number control in polytene chromosomes is relevant to replication in mitotically dividing cells. The abstract implies a general mechanism, which has not been shown.

3) Some improvement in how the paper is written, as outlined by reviewer 2, are suggested.

*Reviewer #1 (Recommendations for the authors):*

1. My main concern about this manuscript is that, as of now, there is little to no evidence that Mod(Mdg4) is involved in under-replication. The copy number measurements that are in that paper are very minor when compared to the SuUR mutant or a Rif1 mutant that fully suppresses under-replication. I understand that they are 'statistically significant' but it is common with genomic approaches that small differences driven by lots of data points are statistically significant, but since many of these changes are a few percent different than the wild type, I'm not sure how biologically significant they are. While the authors provide some ideas of why they don't observe a large difference in under-replication, as of now it is equally likely that Mod(Mdg4) simply has no role in promoting under-replication and the minor changes in copy number observed are indirect effects of the Mod(Mdg4) mutant. For example, have the authors ever measured the changes in transcript levels associated with the loss of Mod(Mdg4) function in salivary glands? Regardless, there are other approaches they could use to circumvent the lethality of the Mod(Mdg4)m9 mutant. For example, what is the under-replication phenotype of the Mod(Mdg4)u1 mutant? Have the authors used a salivary gland-specific drive like fkh to drive RNAi against Mod(Mdg4) instead of using a mutant? Again, given the focus of SUMM4 controlling under-replication, it is essential that the authors provide compelling evidence one way or another. While I appreciate the approach they used to look at under-replication from heterozygous SuUR mutant mothers, it is difficult to compare two different proteins given the difference in protein and RNA levels that are stockpiled for individual genes and differences in protein stability. It's really an apples-to-oranges comparison.

2. In both the introduction (ln 62) and discussion (ln 434), the authors state the biochemical composition of replisomes is identical in polyploid and diploid cells (Zielke et al. 2013). The is absolutely no data in this reference to support this claim. In fact, this reviewer couldn't even find the term replisome in that review article. I find that this is an egregious and misleading overstatement that is not supported by any current literature.

3. Ln 66-67. This is not entirely true. SuUR mutants only partially restore copy number in UR regions – even the IH regions.

4. Figure 3A: It would have been helpful to show a zoom-in with the split channels where half of the chromosome is in red and the other half in green. This would allow the readers to better determine the extent of overlap. Also, it would be nice to swap out green/red for another color pairing.

5. Figure 4A: You point out distinct changes to the wing morphology upon SuUR mutation (red and black arrows). Does that suggest SuUR contributes to insulator function independently of the SUMM4 complex?

6. Figure 4C: What is the result for the double mutant?

7. Ln 268: Reference should be: Sher et al. 2012 and Yarosh and Spradling 2014.

8. Ln 385-388: I don't know how this statement is supported by this data. This is a snapshot from cultured cells. The IF for Mod(Mdg4)-dependent binding is not a high resolution to support this claim.

9. Ln401-406: Trying to understand your model, but why would some SUMM4 complexes be bypassed while others are 'impenetrable'?

10. Ln 409: The authors should perform a proper genome-wide analysis to determine if Su(Hw) binding sites are, in fact, enriched at the boundaries of IH regions. Right now it's just a snapshot of a few UR domains (and is not convincing).

11. There were a few spots in the discussion where it seemed that primary data was being referenced for the first time. It should be moved to the Results section. (e.g. ln 408-409; 413-414)

12. A suggestion for making this easier to read. There is a lot of data in this paper and I'm not sure why the authors have only four main figures. It would have been nice if a few of these figures were not so long. Also, your model figure is buried as Figure 4D/4F. Having a figure that is devoted to the model would have been helpful, or at least try not to bury the model figure.

*Reviewer #2 (Recommendations for the authors):*

1. Too many acronyms, some of which are unnecessary and confusing (e.g. IH, PH, UR). Please spell out the words.

2. Some acronyms are not even defined, as far as I can tell. SUMM4: what does that stand for? Why not call the complex Mdg4-SUUR complex? What does DIA/SWATH stand for?

3. Development of MERCI is not completely clear and expects the reader to be familiar with MS jargon. In particular, in lines 97-101 it was not clear how use of the recombinant SUUR ion library helps in tracking SUUR in the first chromatographic step, phosphocellulose.

4. Line 240. There should be a citation for the physical association reported between Mdg4 and Su(Hw). Perhaps Gause et al. MCB 2001 or another report in the literature.

5. There should be a full paragraph in the discussion that describes in words the authors' mechanistic interpretation of their data to accompany the models in Figure 4D, E, F.

6. Line 438……….'two-stroke mechanism'. Do authors mean two-stage?

7. In describing the effects of mutants, it would be useful to inform readers, from the outset, that in general, many *Drosophila* mutants, by virtue of the maternal contribution from the heterozygous parent, do not reveal their phenotypes until the maternal product has been diluted or degraded, usually in the late larval instar stage. Saying this early in the text will help to understand the weaker effects of some mutants.

---

## [Author Response]

Essential revisions:1) The authors should address the comments of reviewer 1 about the role in controlling the replication timing and under-replication that occurs in polytene chromosomes. Some approaches are suggested.

In the revised manuscript, we have addressed all critical points made by Reviewer 1. Please see our point-by-point responses to his or her comments below.

2) The generality of the proposed mechanism for control of replication timing needs to be discussed, particularly whether this potential mechanism of copy number control in polytene chromosomes is relevant to replication in mitotically dividing cells. The abstract implies a general mechanism, which has not been shown.

We agree that the relevance of under-replication phenomenon to the establishment of late replication in dividing cells has only been established based on circumstantial evidence. In the revised manuscript, we expand the explanation of this relationship (lines 63-65) and discuss limitations of the endoreplication model as applied to understanding of DNA replication timing in a normal cell cycle (lines 476-481). We also modified the Title and Abstract to soften our conclusions and to focus them on late/delayed replication rather than replication timing in general (lines 1, 25-27, 39-41).

3) Some improvement in how the paper is written, as outlined by reviewer 2, are suggested.

We are grateful to the Reviewer for suggestions for improving the clarity of our manuscript. We have revised it accordingly. Please see our point-by-point responses to the comments by Reviewer 2 below.

Reviewer #1 (Recommendations for the authors):1. My main concern about this manuscript is that, as of now, there is little to no evidence that Mod(Mdg4) is involved in under-replication. The copy number measurements that are in that paper are very minor when compared to the SuUR mutant or a Rif1 mutant that fully suppresses under-replication. I understand that they are 'statistically significant' but it is common with genomic approaches that small differences driven by lots of data points are statistically significant, but since many of these changes are a few percent different than the wild type, I'm not sure how biologically significant they are. While the authors provide some ideas of why they don't observe a large difference in under-replication, as of now it is equally likely that Mod(Mdg4) simply has no role in promoting under-replication and the minor changes in copy number observed are indirect effects of the Mod(Mdg4) mutant. For example, have the authors ever measured the changes in transcript levels associated with the loss of Mod(Mdg4) function in salivary glands? Regardless, there are other approaches they could use to circumvent the lethality of the Mod(Mdg4)m9 mutant. For example, what is the under-replication phenotype of the Mod(Mdg4)u1 mutant? Have the authors used a salivary gland-specific drive like fkh to drive RNAi against Mod(Mdg4) instead of using a mutant? Again, given the focus of SUMM4 controlling under-replication, it is essential that the authors provide compelling evidence one way or another. While I appreciate the approach they used to look at under-replication from heterozygous SuUR mutant mothers, it is difficult to compare two different proteins given the difference in protein and RNA levels that are stockpiled for individual genes and differences in protein stability. It's really an apples-to-oranges comparison.

Contrary to the statement by the Reviewer, the recovery of DNA under-replication in polytene salivary glands of *mod(mdg4)* mutant is not “a few percent” but 26% on average (see Table 1). As indicated by this Reviewer, *SuUR* mutation does not fully restore the DNA copy numbers in polytene chromosomes either. Rather, it relieves the extent of under-replication in a wide range from 10% (12D1-E1 region) to ~100%, with an average of 78% (Table 1). Therefore, the suppression of under-replication by *mod(mdg4)* mutation, although on average ~3-fold weaker, cannot be considered “very minor” or insignificant in comparison to that by *SuUR*.

We agree that we did not formally exclude an indirect effect of *mod(mdg4)* mutation on under-replication, which may, for instance, be mediated by transcriptional regulation of a downstream target(s). However, the preponderance of evidence (biochemical and cytological) strongly suggests that the effect depends on SUUR. Specifically, in *mod(mdg4)* mutant polytene chromosomes, we observe a near absence of SUUR in its normal euchromatic loci during early endo-S and a drastically reduced abundance in intercalary heterochromatin during late endo-S, which is comparable to the occupancy loss observed in the *SuUR[ES]* amorphic mutant (Figure 4B and Figure 4—figure supplement 3B). Since Mod(Mdg4) is required for a normal loading of SUUR in chromatin, and SUUR is required for the establishment of under-replication, *mod(mdg4)* mutation has to directly suppress under-replication in a SUUR-dependent fashion.

We have decided to use for our genome-wide analyses a null allele of *mod(mdg4)*, *mod(mdg4)[m9]*, as opposed to the *[u1]* allele because of the potential antimorphic effects in the latter (Figure 4C and lines 255-260). In fact, this allele would not alleviate but, rather, exacerbate the possible issue with an indirect effect of *mod(mdg4)* on under-replication (see above). However, to satisfy in part this criticism by the Reviewer, we have recently analyzed the suppression of under-replication by *mod(mdg4)[u1]* in one well-characterized under-replicated genomic locus (75B11-D2), cytologically and by quantitative real-time PCR. As anticipated and likely because there is no maternal contribution of the full-length Mod(Mdg4)-67.2 from homozygous *mod(mdg4)[u1]* mothers, the effect was quantitatively higher than that in the homozygous *mod(mdg4)[m9]* mutants produced by *inter se* crosses of heterozygous parents. These data are included in Figure 7D of the revised manuscript (see also lines 339-342).

Finally, RNAi-dependent knockdown of *mod(mdg4)* proposed by the Reviewer as a replacement for the classical mutant alleles would not solve the problem of maternal loading because of a similar (and, potentially, slower) gradual decrease of the dose of functional protein levels. Thus, Mod(Mdg4) expression would not be completely abrogated prior to the initiation of endoreplication cycles that start during late embryonic development.

We believe that since *mod(mdg4)* is required for viability, whereas *SuUR* is not, a direct comparison of under-replication in homozygous nulls produced from crosses of heterozygous parents (Figure 7C and D, *cf mod(mdg4)[m9]* and zygotic *SuUR[ES]*) provides the best available control. In other words, it is not an “apple-to-oranges” comparison but a comparison of zygotic functions of the two genes in endoreplication control.

2. In both the introduction (ln 62) and discussion (ln 434), the authors state the biochemical composition of replisomes is identical in polyploid and diploid cells (Zielke et al. 2013). The is absolutely no data in this reference to support this claim. In fact, this reviewer couldn't even find the term replisome in that review article. I find that this is an egregious and misleading overstatement that is not supported by any current literature.

We agree that the way that the reference (Zielke et al. 2013) was placed at the end of the sentences gave a wrong impression that it characterized composition of the replisome. Our intention was to indicate the differences between the cell cycle regulation in mitotic and endoreplicating cells. In contrast, the enzymatic machinery of the replisome has indeed not been the focus of studies in endoreplication, since, by default, both types of cells likely have to utilize the same enzymatic machinery (DNA polymerases, helicases, etc). In the revised manuscript, we moved the reference to an appropriate position in the middle of sentences. We also toned down the statement about the replisome biochemical composition to reflect the current state of knowledge (lines 63-65, 476-481).

3. Ln 66-67. This is not entirely true. SuUR mutants only partially restore copy number in UR regions – even the IH regions.

We thank the Reviewer for this correction. The text has been modified accordingly (lines 69-71).

4. Figure 3A: It would have been helpful to show a zoom-in with the split channels where half of the chromosome is in red and the other half in green. This would allow the readers to better determine the extent of overlap. Also, it would be nice to swap out green/red for another color pairing.

As suggested by the Reviewer, split images of polytene chromosomes are only useful for zoomed-in short polytene chromosome stretches where a near-perfect co-localization is observed. They are less beneficial for partially overlapping distribution patterns as is the case of SUUR and Mod(Mdg4)-67.2. Thus, in Figure 4A (old Figure 3A) and accompanying figures (Figure 4—figure supplement 1A), we were trying to convey a more global picture of chromosome-wide (partial) co-localization, whereas co-localization in a narrow genomic region would be less convincing. Furthermore, we believe that our computational analyses of the respective SUUR- and Mod(Mdg4)-67.2-positive polytene images (Figure 4—figure supplement 1B-D) provide an additional layer of evidence for this partial co-localization. Finally, according to the Reviewer’s recommendation, we now include an image with swapped green and red channels (Figure 4A).

5. Figure 4A: You point out distinct changes to the wing morphology upon SuUR mutation (red and black arrows). Does that suggest SuUR contributes to insulator function independently of the SUMM4 complex?

We thank the Reviewer for this comment. The proper statement would be that SUUR may contribute to the insulator function in the absence of Mod(Mdg4)-67.2, which is the only splice form of Mod(Mdg4) that is functionally disrupted in the *[u1]* allele. This statement has been inserted in the revised manuscript (lines 277-280). In fact, it is far from surprising. As we indicated (lines 160-162), our biochemical and mass-spectroscopy data do not exclude a possibility that SUUR forms a complex with other, minor splice form(s) of Mod(Mdg4). This putative SUMM4-like complex(es) may additionally contribute to the activity of *gypsy* insulators in the *ct[6]* allele when the full-length Mod(Mdg4)-67.2 is not expressed. Alternatively, SUUR may have a completely Mod(Mdg4)-independent function at insulators, where it may be recruited by a different mechanism. For instance, in polytene IF images, we observe a small number of euchromatic SUUR-positive loci that are free of Mod(Mdg4)-67.2 (Figure 4A, Figure 4—figure supplement 1B). Also, in the absence of Mod(Mdg4), SUUR occupancy recovers (weakly/partially) in a cell cycle-dependent manner (Figure 4—figure supplement 3B). However, our data do not allow to discriminate between these two alternatives (additional SUMM4-like complexes and Mod(Mdg4)-independent function of SUUR).

6. Figure 4C: What is the result for the double mutant?

In contrast to the *ct[6]* that is localized in the X chromosome, *P{SUPor-P}* insertions that we used in Figure 5C are positioned on the second chromosome. Thus, to generate double homozygous *P{SUPor-P}*; *SuUR, mod(mdg4)* flies, we had to perform crosses of heterozygotes with double (*CyO* and *TM6B, Tb*) balancers. The double-balancer genetic crosses frequently lead to genomic instability. This instability is further aggravated by the pericentric localization of *P{SUPor-P}* insertions, which are not strongly balanced by *CyO*, and by homozygous *SuUR, mod(mdg4)* being a very weak allele. On several attempts, we failed to produce the double homozygous *P{SUPor-P}*; *SuUR, mod(mdg4)* line, although we have succeeded with *P{SUPorP}*; *SuUR* and *P{SUPor-P}*; *mod(mdg4)* flies. We decided to give up on further attempts, since the *white* reporter is almost completely silenced in *P{SUPor-P}*; *mod(mdg4)*, and any additional repression that might be produced by *SuUR* mutation would be (i) difficult to discern and (ii) peripheral to the main conclusions of the paper.

7. Ln 268: Reference should be: Sher et al. 2012 and Yarosh and Spradling 2014.

We have inserted these references (line 298).

8. Ln 385-388: I don't know how this statement is supported by this data. This is a snapshot from cultured cells. The IF for Mod(Mdg4)-dependent binding is not a high resolution to support this claim.

We entirely agree that “propagation” was an inappropriate word to describe the DamID data from (Filion et al. 2010), because it in part implied a dynamic nature of their observations. We also agree that this part of the manuscript sounded as an over-statement. We have modified and toned down our speculation, specifically replacing “propagation” with “occupancy” and “supported by” with “consistent with” (lines 426-431).

9. Ln401-406: Trying to understand your model, but why would some SUMM4 complexes be bypassed while others are 'impenetrable'?

There are numerous potential mechanisms that might explain the relative “strength” of distinct insulator barriers to the replisome:

biophysical (different affinities of protein-DNA binding at particular insulator sequences);biochemical (contributions from additional protein factors, such as Rif1, which may be recruited to a subset of loci independently of SUMM4);epigenetic (variable histone modification states that stabilize/destabilize SUMM4 interactions with chromatin or affect the efficiency of DNA replication);thermodynamic (clustering of multiple insulator sites, as observed frequently at the boundaries of under-replicated domains).

However, without experimental evidence, we deemed these obvious possibilities to be too speculative to discuss in the manuscript.

10. Ln 409: The authors should perform a proper genome-wide analysis to determine if Su(Hw) binding sites are, in fact, enriched at the boundaries of IH regions. Right now it's just a snapshot of a few UR domains (and is not convincing).

These genome-wide analyses have been performed by Hidden Markov modeling in (Khoroshko et al. 2016) as discussed later in the same paragraph (lines 457-460).

11. There were a few spots in the discussion where it seemed that primary data was being referenced for the first time. It should be moved to the Results section. (e.g. ln 408-409; 413-414)

In agreement with the Reviewer’s suggestion, we inserted the statement about clustered Su(Hw) sites in the Results section (lines 309-311).

12. A suggestion for making this easier to read. There is a lot of data in this paper and I'm not sure why the authors have only four main figures. It would have been nice if a few of these figures were not so long. Also, your model figure is buried as Figure 4D/4F. Having a figure that is devoted to the model would have been helpful, or at least try not to bury the model figure.

As proposed by the Reviewer, we split several figures into smaller units, thus increasing the number of main figures from four to seven in the revised manuscript. The model panels (previously, Figure 4D-F) were segregated in a separate Figure 6 and discussed separately (lines 421-426).

Reviewer #2 (Recommendations for the authors):1. Too many acronyms, some of which are unnecessary and confusing (e.g. IH, PH, UR). Please spell out the words.

In the revised manuscript, we spelled out the majority of abbreviations.

2. Some acronyms are not even defined, as far as I can tell. SUMM4: what does that stand for? Why not call the complex Mdg4-SUUR complex? What does DIA/SWATH stand for?

In the revised manuscript, we appropriately defined SUMM4 (lines 146-148) and DIA/SWATH (lines 101-105).

3. Development of MERCI is not completely clear and expects the reader to be familiar with MS jargon. In particular, in lines 97-101 it was not clear how use of the recombinant SUUR ion library helps in tracking SUUR in the first chromatographic step, phosphocellulose.

We agree that understanding of the MERCI approach requires some familiarity with the technical concepts of mass-spectrometry. To help more naïve readers, in the revised manuscript, we expanded some explanations (*e.g*., lines 99-106). Also, we included a clearer justification for using the recombinant SUUR ion library (ILR) to quantitate data from the phosphocellulose SWATH (lines 108-113).

4. Line 240. There should be a citation for the physical association reported between Mdg4 and Su(Hw). Perhaps Gause et al. MCB 2001 or another report in the literature.

We thank the Reviewer for pointing out this omission. We included the reference in the revised manuscript (line 264).

5. There should be a full paragraph in the discussion that describes in words the authors' mechanistic interpretation of their data to accompany the models in Figure 4D, E, F.

As suggested by Reviewer 1, we split several figures into smaller units and segregated the model panels in a separate Figure 6. We also inserted a separate statement in the Discussion to provide a better explanation of our mechanistic models (lines 421-426).

6. Line 438……….'two-stroke mechanism'. Do authors mean two-stage?

In the revised manuscript, we replaced “two-stroke” with “two-component” (line 484).

7. In describing the effects of mutants, it would be useful to inform readers, from the outset, that in general, many *Drosophila* mutants, by virtue of the maternal contribution from the heterozygous parent, do not reveal their phenotypes until the maternal product has been diluted or degraded, usually in the late larval instar stage. Saying this early in the text will help to understand the weaker effects of some mutants.

In accordance with the Reviewer’s suggestion, we have added this statement in the revised manuscript (lines 229-234).